# Regulation of lipid saturation without sensing membrane fluidity

Stephanie Ballweg[1,2], Erdinc Sezgin [3], Milka Doktorova[4], Roberto Covino [5], John Reinhard[1,2], Dorith Wunnicke[6], Inga Hänelt [6], Ilya Levental [4], Gerhard Hummer [5,7] & Robert Ernst [1,2]*

Cells maintain membrane fluidity by regulating lipid saturation, but the molecular mechanisms of this homeoviscous adaptation remain poorly understood. We have reconstituted the core machinery for regulating lipid saturation in baker's yeast to study its molecular mechanism. By combining molecular dynamics simulations with experiments, we uncover a remarkable sensitivity of the transcriptional regulator Mga2 to the abundance, position, and configuration of double bonds in lipid acyl chains, and provide insights into the molecular rules of membrane adaptation. Our data challenge the prevailing hypothesis that membrane fluidity serves as the measured variable for regulating lipid saturation. Rather, we show that Mga2 senses the molecular lipid-packing density in a defined region of the membrane. Our findings suggest that membrane property sensors have evolved remarkable sensitivities to highly specific aspects of membrane structure and dynamics, thus paving the way toward the development of genetically encoded reporters for such properties in the future.

[1] Medical Biochemistry and Molecular Biology, Medical Faculty, Saarland University, Kirrberger Strasse 100, Building 61.4, 66421 Homburg, Germany. [2] PZMS, Center for Molecular Signaling (PZMS), Medical Faculty, Saarland University, 66421 Homburg, Germany. [3] MRC Human Immunology Unit, MRC Weatherall Institute of Molecular Medicine, University of Oxford, Oxford, UK. [4] Department of Integrative Biology and Pharmacology, McGovern Medical School at the University of Texas Health Science Center, Houston, Texas, USA. [5] Department of Theoretical Biophysics, Max Planck Institute of Biophysics, Max-von-Laue-Strasse 3, 60438 Frankfurt, Germany. [6] Institute of Biochemistry, Goethe University Frankfurt, Max-von-Laue-Strasse 9, 60438 Frankfurt, Germany. [7] Institute of Biophysics, Goethe University Frankfurt, 60438 Frankfurt, Germany. *email: robert.ernst@uks.eu

Cellular membranes are complex assemblies of proteins and lipids, which collectively determine physical bilayer properties such as membrane fluidity/viscosity, permeability, and the lateral pressure profile[1–4]. Membrane fluidity determines how easily lipids and proteins can diffuse laterally in the plane of the membrane. We quantify it by measuring the diffusion coefficient of membrane lipids, which is inversely proportional to membrane viscosity. The acyl chain composition of membrane lipids is an important determinant of membrane fluidity and is tightly controlled in bacteria[5–7], fungi[8,9], worms[10,11], flies[12], and vertebrates[13,14]. Saturated lipid acyl chains tend to form non-fluid, tightly packed gel phases at physiological temperatures, whereas unsaturated lipid acyl chains fluidize the bilayer. Poikilothermic organisms that cannot control their body temperature must adjust their lipid composition during cold stress to maintain membrane functions—a phenomenon referred to as the homeoviscous adaptation[15–17]. Despite recent advances in identifying candidate sensory machineries, it remains largely unknown how they work on the molecular scale and how they are coordinated for maintaining the physicochemical properties of cellular membranes[18,19]. The fact that most, if not all, membrane properties are interdependent is a key challenge for this emerging field. How do cells, e.g., balance the need for maintaining membrane fluidity with the need to maintain organelle-specific lateral pressure profiles[20]? A perturbation of membrane fluidity by genetically targeting fatty acid metabolism, e.g., leads to complex changes throughout the entire lipidome impacting on other bilayer properties, thereby causing endoplasmic reticulum (ER) stress and a disruption of membrane architecture[21–23]. We lack a unifying theory to accurately predict the properties of a membrane given its composition: each component of a complex biological membrane contributes to its collective physicochemical properties in a non-additive and nonlinear manner[3,24]. As an important step towards a unifying membrane theory, we need to identify a set of membrane properties that are both minimally correlated and sufficient to uniquely describe the state of a bilayer. Characterizing naturally occurring membrane property sensors, which may exhibit highly specialized sensitivities to specific membrane properties, holds promise to better understand how cells prioritize the maintenance of such orthogonal membrane properties[19].

Eukaryotic cells use sensor proteins possessing refined mechanisms to monitor physicochemical properties of organellar membranes and to adjust lipid metabolism during stress, metabolic adaptation, and development[10,23,25–30]. These sensor proteins can be categorized into three classes, based on topological considerations[18,19]: Class I sensors interrogate surface properties of cellular membranes, such as the surface charge and molecular packing density as reported for amphipathic lipid-packing sensor (ALPS) motif-containing proteins and other amphipathic helix-containing proteins[31]. Class II sensors perturb and interrogate the hydrophobic core of the bilayer and have been implicated in the regulation of lipid saturation. Class III sensors are transmembrane proteins acting across the bilayer by locally squeezing, stretching, and/or bending the membrane to challenge selective properties such as thickness or bending rigidity[18,19].

The prototypical class II sensor Mga2 is crucial for the regulation of membrane fluidity in baker's yeast[9,25] (Fig. 1a). Its single transmembrane helix (TMH) senses a physicochemical signal in the ER membrane to control a homeostatic response that adjusts membrane lipid saturation via the essential fatty acid cis-Δ9-desaturase Ole1[32–34]. Increased lipid saturation triggers the ubiquitylation of three lysine residues in the cytosolic juxtamembrane region of Mga2 by the E3 ubiquitin ligase Rsp5[35]. This ubiquitylation serves as a signal for the proteasome-dependent processing of the membrane-bound Mga2 precursor (P120) and

the release of a transcriptionally active P90 fragment, which upregulates OLE1 expression[36] (Fig. 1a). This regulated, ubiquitin/proteasome-dependent processing resembles the pathway of ER-associated degradation (ERAD)[37] and was first described for Spt23, a close structural and functional homolog of Mga2[38]. As Ole1 is the only source for the de novo biosynthesis of unsaturated fatty acids (UFAs), its tight regulation is essential for maintaining membrane fluidity in this poikilotherm[9,34].

Molecular dynamics (MD) simulations have revealed a remarkable conformational flexibility of the Mga2 transmembrane region[25]. The TMHs of Mga2 dimerize and rotate against each other, thus forming an ensemble of dimerization interfaces. Importantly, the population of these alternative configurations is affected by the membrane lipid environment: higher proportions of saturated lipid acyl chains stabilize a configuration in which two tryptophan residues (W1042) point toward the dimer interface, whereas higher proportions of unsaturated lipid acyl chains favor a conformation where these residues point away from one another and toward the lipid environment[9,25]. Based on the remarkable correspondence with genetic and biophysical data, we proposed that the membrane-dependent structural dynamics of the TMHs are coupled to the ubiquitylation and activation of Mga2[25]. However, it remained unclear whether the reported, relatively subtle changes in the population of short-lived rotational conformations are sufficient to control a robust cellular response. How can the processing of Mga2 be blocked by an increased proportion of unsaturated lipids in the membrane, if the sensory TMHs still explore their entire conformational space? How is the "noisy" signal from the TMH propagated via disordered regions to the site of ubiquitylation in the juxtamembrane region (Fig. 1b)?

As an important step toward answering these questions, we have designed and isolated a minimal sensor construct based on Mga2 that can both sense and respond: it senses the membrane environment and acquires, depending on the membrane lipid composition, a poly-ubiquitylation label as a signal for its activation via proteasomal processing. After reconstituting this sense-and-response construct in liposomes with defined lipid compositions, we demonstrate a remarkable sensitivity of Mga2 to specific changes in the bilayer composition. We provide evidence for functional coupling between the TMH and the site of ubiquitylation using electron paramagnetic resonance (EPR) and Förster resonance energy transfer (FRET). Our data contradict a central assumption of the theory of homeoviscous adaptation and rule out the possibility that Mga2 acts as a sensor for membrane fluidity. Instead, we propose that Mga2 senses the packing density at the level of the sensory tryptophans (W1042)[25] and thus a small portion of the lateral compressibility profile in the hydrophobic core of the membrane. Analogous to ALPS motifs that recognize lipid-packing defects in the water-membrane interface by inserting hydrophobic residues into the membrane core[39], Mga2 might sense the packing density of hydrogen and carbon atoms in the core of the membrane via the bulky residue W1042. We speculate that this packing-dependent sensing, together with chemical interactions, determines the population of different rotational orientations of the entire TMH of Mga2. Thus, our mechanistic analysis of the membrane lipid saturation sensor Mga2 challenges the common view of membrane fluidity as the critical measured variable in membrane biology.

## Results

**A minimal reporter of membrane lipid saturation**. We proposed that Mga2 uses a rotation-based mechanism to sense membrane lipid saturation[25] (Fig. 1a). However, the sensory TMHs of Mga2 are separated from the site of ubiquitylation by a

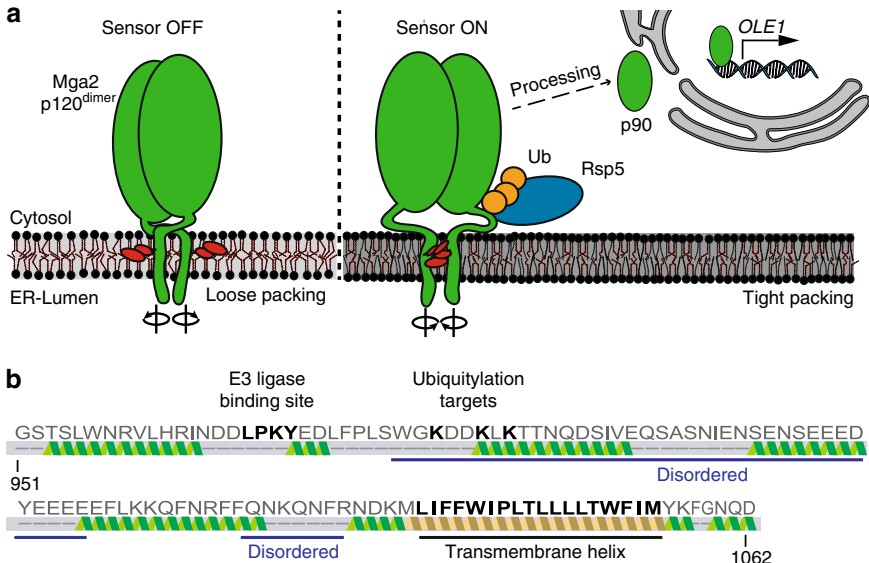

**Fig. 1 The activation of Mga2 is controlled by the ER membrane composition. a** Model of the OLE pathway: the transcription factor Mga2 forms inactive dimers in the ER membrane (Mga2 p120$^{dimer}$) with highly dynamic TMHs exploring alternative rotational orientations. Loose lipid packing (left) caused by unsaturated lipids stabilizes conformations with two sensory tryptophan residues (W1042; red) pointing away from the dimer interface toward the lipid environment. Tight lipid packing (right) stabilizes alternative rotational conformations with the sensory tryptophans facing each other in the dimer interface (right). The E3 ubiquitin ligase Rsp5 is required to ubiquitylate (Ub) Mga2, thereby facilitating the proteolytic processing by the proteasome and the release of transcriptionally active Mga2 (p90). **b** Secondary structure prediction of the juxtamembrane and transmembrane region (residue 951–1062) of Mga2 using Phyre2[65]. Source data are provided as a Source Data file.

predicted disordered loop and ~50 amino acids (Fig. 1b), thereby posing a question of their functional coupling. How can the conformational dynamics of the TMHs control the ubiquitylation of Mga2? To study the coupling of sensing and ubiquitylation in vitro, we have generated a minimal sense-and-response construct ($^{ZIP-MBP}$Mga2$^{950–1062}$) comprising an N-terminal leucine zipper (ZIP) derived from the transcription factor GCN4, the maltose-binding protein (MBP), the juxtamembrane region (950–1036), and the TMH (1037–1058) of Mga2 (950–1062) (Fig. 2a). The N-terminal zipper mimics the IPT (Ig-like, plexin, transcription factor) domain of full-length Mga2 and stabilizes a homo-dimeric state[40], whereas the MBP was used as a purification and solubility tag[25]. The juxtamembrane domain of Mga2 comprises the LPKY motif (Mga2$^{958–961}$) for recruiting the E3 ubiquitin ligase Rsp5[41], three lysine residues K$^{980}$, K$^{983}$, and K$^{985}$ ubiquitylated in vivo[35], and the disordered region linking these motifs to the TMH (Fig. 2a). The construct was recombinantly produced and isolated in the presence of Octyl-β-D-glucopyranoside (OG) using an amylose-coupled affinity matrix and size-exclusion chromatography (SEC) (Fig. 2b and Supplementary Fig. 1a). Expectedly, the N-terminal zipper stabilizes a dimeric form of the sense-and-response construct and supports, at increased concentrations, the formation of higher oligomeric forms as suggested by SEC experiments that also included a zipper-less variant ($^{MBP}$Mga2$^{950–1062}$) as a control (Fig. 2c and Supplementary Fig. 1b, c). We reconstituted the sense-and-response construct in liposomes at molar protein-to-lipid ratios between 1:5000 and 1:15,000. As illustrated by the protein-to-lipid ratio of 1:8000 (Supplementary Fig. 1d), we did not detect any sign of protein aggregation in our preparations using sucrose density gradient centrifugations.

We then tested whether the sense-and-response construct could be ubiquitylated in vitro and adapted a strategy established for the ubiquitylation of substrates of the ERAD machinery[42]. We incubated the proteoliposomes with an ATP-regenerating system,

purified $^{8xHis}$ubiquitin, and yeast cytosol containing all enzymes required for ubiquitylation (Fig. 2d). Subsequent immunoblot analyses revealed a time-dependent ubiquitylation of the sense-and-response construct, which became apparent as a ladder of MBP-positive signals (Fig. 2e). Control experiments validated the specificity of the ubiquitylation reaction: no ubiquitylation was observed, when the Rsp5-binding site (ΔLPKY) was deleted from the sense-and-response construct (Fig. 2e). Furthermore, despite the presence of 50 lysine residues in the entire construct, the substitution of the three lysine residues (3KR) targeted by Rsp5 in vivo[35] was sufficient to prevent the ubiquitylation (Fig. 2e). Notably, these experiments were performed at a relatively high protein-to-lipid ratio of 1:5000, to increase the sensitivity for a potential background ubiquitylation of the control constructs. We conclude that the in vitro ubiquitylation assay is specific, and that the conformational dynamics in the juxtamembrane region are likely to reflect the structural dynamics found in full-length Mga2. Most importantly, this in vitro system also allowed us to test the hypothesis of functional coupling between the sensory TMHs and protein ubiquitylation.

We reconstituted the sense-and-response construct in two distinct membrane environments based on a phosphatidylcholine (PC) matrix but differing in their lipid acyl chain composition. One membrane environment contained 50% unsaturated 18:1 and 50% saturated 16:0 acyl chains (100 mol% 1-palmitoyl-2-oleoyl-*sn*-glycero-3-phosphocholine (POPC); 16:0/18:1), whereas the other was less saturated and contained 75% unsaturated 18:1 and 25% saturated 16:0 acyl chains (50 mol% 1,2-dioleoyl-*sn*-glycero-3-phosphocholine (DOPC); 18:1/18:1, 50 mol% POPC) (Fig. 2f). Notably, this degree of lipid saturation is in the range of the naturally occurring acyl chain compositions reported for baker's yeast cultivated in different conditions[8,21,43,44]. The sense-and-response construct was efficiently ubiquitylated in the more saturated membrane environment (evidenced by a ladder of bands with lower electrophoretic mobility), but not in the

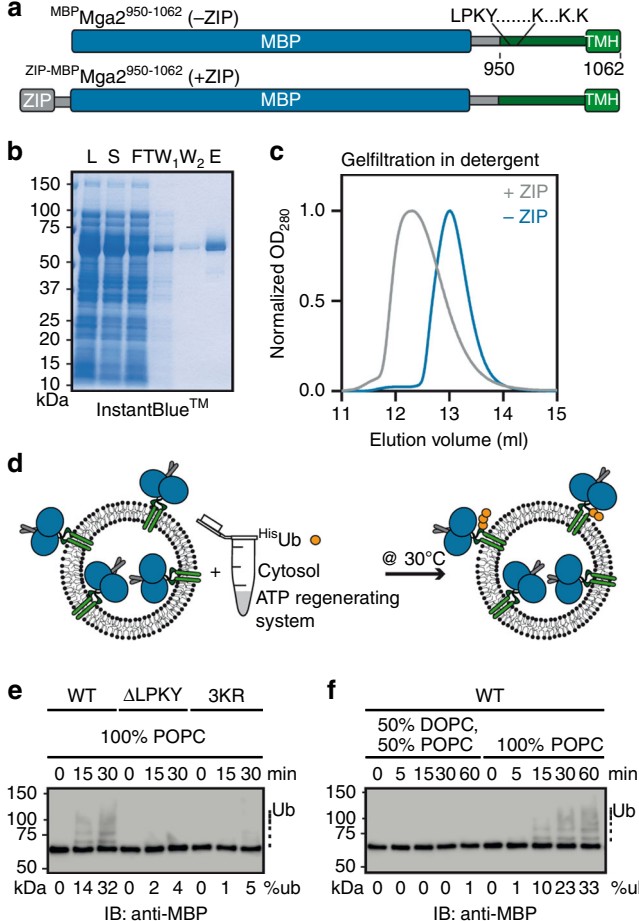

**Fig. 2 An in vitro sense-and-response system for membrane lipid saturation. a** Schematic representation of the sense-and-response constructs. The fusion proteins are composed of the maltose-binding protein (MBP, blue) and Mga2$^{950-1062}$ (green) with the Rsp5-binding site (LPKY), three lysine residues as targets of ubiquitylation (K$^{980}$, K$^{983}$, and K$^{985}$), a predicted disordered juxtamembrane region, and the C-terminal TMH. An optional N-terminal leucine zipper derived from Gcn4 (gray, Gcn4$^{249-281}$) supports dimerization. **b** Isolation of the zipped sense-and-response construct by affinity purification. 0.1 OD units of the lysate (L), soluble (S), flow-through (FT), and two wash fractions (W$_{1,2}$), as well as 1 μg of the eluate were subjected to SDS-PAGE followed by InstantBlue$^{TM}$ staining. The protein was further purified by preparative SEC (Supplementary Fig. 1a). **c** One hundred micrograms in 100 μl of the purified sense-and-response constructs either with (+ZIP) or without zipper (−ZIP) were loaded onto a Superdex 200 10/300 Increase column (void volume 8.8 ml). **d** Schematic representation of the in vitro ubiquitylation assay. Proteoliposomes containing $^{ZIP-MBP}$Mga2$^{950-1062}$ were mixed with $^{8xHis}$Ubiquitin ($^{His}$Ub), an ATP-regenerating system, and cytosol prepared from wild-type yeasts to facilitate Mga2 ubiquitylation at 30 °C. **e** The reaction was performed with the $^{ZIP-MBP}$Mga2$^{950-1062}$ wild-type (WT) construct, a variant lacking the Rsp5-binding site (ΔLPKY), and a variant with arginine residues instead of the lysine residues K$^{980}$, K$^{983}$, and K$^{985}$ (3KR), thus lacking the target residues of ubiquitylation. These variants were reconstituted in liposomes composed of 100 mol% POPC at protein-to-lipid ratio of 1:5000. After the indicated times, the reactions were stopped using sample buffer and were subjected to SDS-PAGE. For analysis, an immunoblot using anti-MBP antibodies was performed. **f** Ubiquitylation reactions were performed as in **e** with the WT sense-and-response construct reconstituted in the indicated lipid environments at a molar protein-to-lipid ratio of 1:5000. Source data are provided as a Source Data file.

unsaturated one (Fig. 2f and Supplementary Fig. 1e). This observation highlights the remarkable sensitivity of this class II membrane property sensor and provides strong evidence for a functional coupling between the TMHs and the site of ubiquitylation.

**An in vitro strategy to reconstitute membrane lipid sensing.** To detect changes of the conformational dynamics in the juxta-membrane region, we established an in vitro FRET assay. We hypothesized that the average distance between the binding site of the E3 ligase Rsp5 (LPKY) and a lysine residue targeted by Rsp5 may be affected by changes in the membrane lipid environment due to a functional coupling with the TMH. We thus generated a donor construct labeled with Atto488 at the position of a lysine residue (K983$^{D}$) targeted by the ubiquitylation machinery and an acceptor construct labeled with Atto590 within the Rsp5 recognition site (K969$^{A}$) (Förster radius of 59 Å) (Fig. 3a). Notably, the required amino acid substitutions to cysteine at the positions of labeling did not interfere with the activation of full-length Mga2 in vivo (Supplementary Fig. 2a). The individually isolated donor (K983$^{D}$) and acceptor (K969$^{A}$) constructs exhibited only negligible fluorescence emission at 614 nm in detergent solution upon donor excitation at 488 nm (Fig. 3b). However, a significant emission at 614 nm (from here on referred to as FRET signal) was detectable upon mixing the donor and acceptor constructs (K983$^{D}$ + K969$^{A}$) (Fig. 3b). Notably, a direct excitation of the acceptor at 590 nm (Supplementary Fig. 2b) resulted in equal fluorescence intensities at 614 nm for both K983$^{D}$ + K969$^{A}$ and K969$^{A,only}$ samples, but no emission for the K983$^{D,only}$ sample. The normalized FRET signal of the K983$^{D}$ + K969$^{A}$ reporter was concentration-dependent in detergent solution (Fig. 3c), thereby suggesting a dynamic equilibrium between monomeric and oligomeric species (presumably dimers) of the labeled sense-and-response construct. To validate this interpretation and to rule out the possibility that the FRET signal was predominantly caused by FRET between stable K983$^{D}$-K983$^{D}$ and K969$^{A}$-K969$^{A}$ dimers bumping into each other, we performed competition experiments. We found that the ratiometric FRET efficiency of the K983$^{D}$ + K969$^{A}$ reporter was substantially reduced upon titrating it with an unlabeled sense-and-response construct containing an N-terminal leucine zipper (Fig. 3d). However, it remained unaffected upon titration with an unlabeled construct lacking a zipper (Fig. 3d). This indicates (i) that the zipper centrally contributes to the stability of the dimer, (ii) that individual protomers readily exchange in detergent solution, and (iii) that the FRET signal is mainly due to K983$^{D}$-K969$^{A}$ heterooligomers. In fact, additional titration experiments with the K969$^{A}$ acceptor revealed that the observed FRET efficiency is a linear function of the molar fraction of the acceptor (Supplementary Fig. 2c, d), thereby indicating that the FRET signal is indeed caused by dimers[45].

Next, we studied the structural dynamics of the sense-and-response construct in liposomes using the FRET reporter. To this end, we reconstituted K983$^{D,only}$ and the premixed K983$^{D}$ + K969$^{A}$ pair in liposomes of defined lipid compositions and recorded fluorescence spectra (Fig. 3e–g). We used a low protein-to-lipid ratio of 1:8000 in these experiments to minimize the contribution of unspecific proximity FRET to the overall signal[45]. We observed a significant FRET signal for the K983$^{D}$-K969$^{A}$ reporter reconstituted in a POPC bilayer (Fig. 3e) evidenced by a decreased donor fluorescence and an increased acceptor emission at 614 nm compared with the K983$^{D,only}$ sample (Fig. 3e). Using this FRET assay, we then studied the impact of the lipid acyl chain composition on the structural dynamics of the juxtamembrane region. The lowest FRET efficiency was observed in a DOPC bilayer containing 100% unsaturated acyl chains

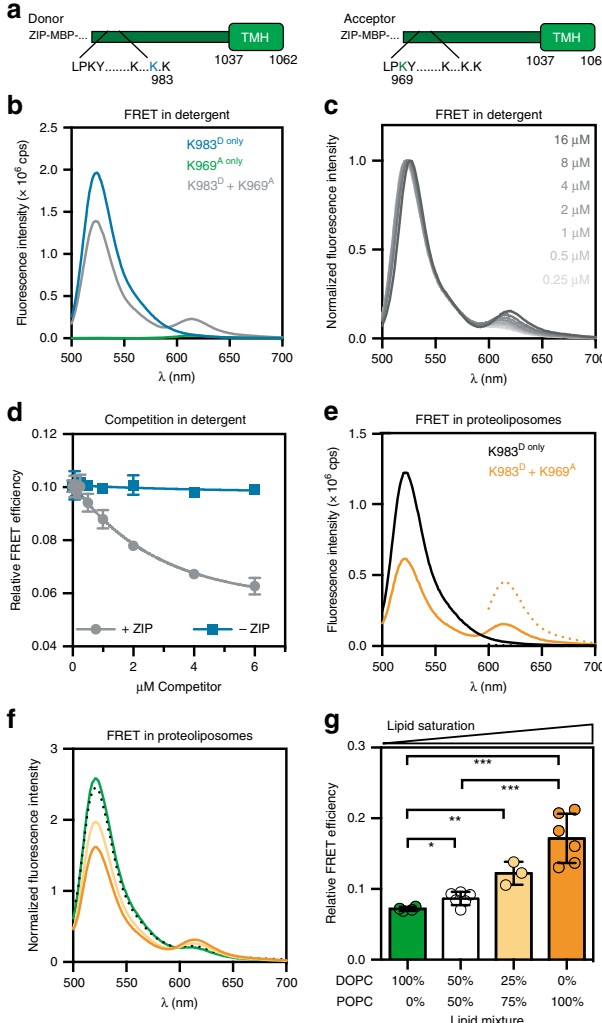

**Fig. 3 FRET reveals membrane-dependent conformational changes in the sense-and-response construct. a** Representation of constructs. The Atto488 dye is linked to K983C at a position, which is ubiquitylated by Rsp5 in vivo. The Atto590 dye was linked to K969C in the Rsp5-binding site. **b** Fluorescence emission spectra reveal FRET in detergent solution. Each construct (2 μM) were used to record fluorescence emission spectra (ex: 488 nm, em: 500–700 nm) of the donor (K983$^D$ only), acceptor (K969$^A$ only), and the combined (K983$^D$ + K969$^A$) FRET pair. **c** Fluorescence emission spectra were recorded for serial dilutions K983$^D$ + K969$^A$ in detergent solution as in **b**. The spectra were normalized to the maximal intensity at the donor emission. **d** Zipped donor (2 μM) and acceptor (2 μM) pairs were mixed and incubated in detergent solution for 10 min, to allow for protomer exchange and equilibration. This sample was titrated with an unlabeled competitor either with (+ZIP) or without zipper domain (−ZIP). Emission spectra were recorded as in **b**. The relative FRET efficiency was determined from the acceptor-to-donor intensity ratio and plotted as mean ± SD from two independent experiments. **e** Emission spectra indicate energy transfer within the membrane-reconstituted, dimeric sense-and-response construct. The donor construct was premixed either with an unlabeled (K983$^D$ only) or a labeled acceptor construct (K983$^D$ + K969$^A$) prior to the reconstitution in POPC liposomes at a protein-to-lipid ratio of 1:8000. Fluorescence emission spectra (em: 500–700 nm) upon donor excitation (ex: 488 nm; solid line) and acceptor excitation (ex: 590 nm; dotted line) are plotted. **f** Donor (K983$^D$) and acceptor (K969$^A$) were mixed equimolarly and were incubated in detergent solution prior to a reconstitution in indicated lipid environments. Emission spectra were recorded as in **e** and were normalized to the maximal acceptor emission after direct acceptor excitation (ex: 590 nm). **g** The relative FRET efficiency was derived from the fluorescence spectra in **f** and plotted as mean ± SD of at least three independent reconstitutions ($n_{DOPC} = 4$; $n_{1:1. (POPC:DOPC)} = 6$; $n_{3:1(POPC:DOPC)} = 3$; $n_{POPC} = 6$). A two-tailed, unpaired t-test was performed to test for statistical significance (*$p < 0.05$, **$p < 0.01$, ***$p < 0.001$). Source data are provided as a Source Data file.

(Fig. 3f, g). At higher proportions of saturated lipid acyl chains in the bilayer, the FRET efficiency increased. These data demonstrate that the acyl chain composition in the hydrophobic core of the membrane imposes structural changes to regions outside the membrane, which have been implicated in signal propagation[35,41]. Our data establish an intricate functional and structural coupling between the TMH regions and the sites of ubiquitylation.

**The Mga2-based reporter does not sense membrane fluidity.** We performed extensive MD simulations illustrating that bilayer properties such as the average area per lipid (Supplementary Fig. 3a), the membrane thickness (Supplementary Fig. 3b), the lateral pressure profile (Supplementary Fig. 3c), and the lipid acyl chain order (Supplementary Fig. 3d) are determined both by the lipid headgroups and the lipid acyl chains.

The remarkable sensitivity of Mga2 to lipid saturation raises the question of whether it is based on membrane fluidity. To test this hypothesis, we first measured the diffusion coefficients of fluorescent lipid analogs (0.01 mol% Atto488-DPPE (Fig. 4a) and 0.01 mol% Abberior Star Red-PEG Cholesterol) (Supplementary Fig. 4a) in giant unilaminar vesicles with different lipid compositions via confocal point fluorescence correlation spectroscopy (FCS). Expectedly, the membrane fluidity decreases slightly with the proportion of saturated lipid acyl chains (from 0% saturated acyl chains for DOPC to 50% for POPC) as evidenced

by decreased diffusion coefficients of the labeled lipids (Fig. 4a and Supplementary Fig. 4a). Notably, this lipid diffusion coefficient is roughly an order of magnitude higher than in cellular membranes[46] and is ~3- to 5-fold lower than in isolated giant plasma membrane vesicles[47]. Previous reports have identified a central contribution of phosphatidylethanolamine (PE) to membrane fluidity in cells and in vitro[48]. Consistently, we see that a lipid bilayer containing 40 mol% PE (Fig. 4a and Supplementary Fig. 4a gray symbols) is less fluid than a bilayer composed only of PC lipids, despite an identical acyl chain composition (Fig. 4a and Supplementary Fig. 4a white symbols). Intriguingly, the anisotropy of the membrane-probe 1,6-diphenyl 1,3,5-hexatriene (DPH) is barely affected by such changes of the bilayer composition over a broad range of temperatures (Supplementary Fig. 4b). This indicates that subtle changes of membrane fluidity are better detected by analyzing lipid diffusion rather than by the anisotropy of the smaller DPH probe. We further characterized this set of lipid compositions using C-Laurdan spectroscopy, which reports on water penetration into the lipid bilayer[49]. A low degree of water penetration increases the generalized polarization (GP) of C-Laurdan and indicates tighter lipid packing in the lipid headgroup region. Expectedly, we found that lipid packing increases with lipid saturation and upon including PE lipids into the bilayer (Fig. 4b and Supplementary Figs. 3a and 4c).

If Mga2 directly sensed membrane viscosity, the fluidity of the bilayer should dominate the structural dynamics of both the sensory TMHs and at the site of ubiquitylation (Fig. 3). Most importantly, the membrane viscosity should then also control the ubiquitylation of the sense-and-response construct (Fig. 2). We

tested these predictions by using up to 40 mol% of PE in the lipid bilayer to perturb membrane fluidity without changing the composition of its lipid acyl chains (Fig. 4). Notably, PE does not only affect membrane fluidity but also other membrane properties such as lipid packing in the headgroup region (Fig. 4b), the area per lipid, the membrane thickness, and the lateral pressure profile as highlighted in MD simulations (Supplementary Fig. 3).

Nevertheless, EPR spectroscopy indicated that different proportions of PE in the bilayer do not substantially perturb the conformational dynamics in the sensory TMH of Mga2 (Fig. 4c and Supplementary Fig. 4d). We took advantage of a previously established minimal sensor construct, which comprises the TMH of Mga2 (residues 1029–1062) fused to MBP[25]. Using methanethiosulfonate (MTS) spin labels installed at the position of W1042 in the TMH and continuous wave EPR spectroscopy, we had previously observed a significant impact of lipid saturation on the average interspin distance[25]. The relatively high protein-to-lipid ratio of 1:500 required for these EPR experiments render the resulting proteoliposomes a bit more similar to cellular membranes, which exhibit protein-to-lipid ratios of ~1:80[18]. We show that the inclusion of 20% or 40 mol% of PE in the proteoliposomes has no discernable impact on the resulting EPR spectra (Supplementary Fig. 4d) and the semi-quantitative value for average interspin proximity (the $I_{Lf}/I_{Mf}$ ratio) (Fig. 4c), despite its impact on membrane fluidity (Fig. 4a and Supplementary Fig. 4a, e) and other membrane properties (Fig. 4b and Supplementary Figs. 3 and 4c). Hence, it is unlikely that the previously reported impact of lipid saturation on the structural dynamics of the TMH[25] is due to a decreased membrane fluidity (Fig. 4a).

Our FRET reporter (K983[D] + K969[A]) enabled us to reveal the role of membrane viscosity on the structural dynamics in the region of ubiquitylation (Fig. 3). We determined the average diameter of proteoliposomes containing the FRET reporter by dynamic light scattering (Malvern Zetasizer Nano S90) (Supplementary Fig. 4f). It was slightly larger (~140 nm) for proteoliposomes containing 40 mol% PE than for those generated exclusively from PC lipids (100 mol% DOPC, 50 mol% DOPC and 50 mol% POPC, and 100 mol% POPC) with average diameters between ~60 nm and ~75 nm. Although membrane curvature can have a significant impact on most membrane properties including the lateral pressure profile and lipid packing, we doubt that the curvature in proteoliposomes with diameters > 60 nm manifests substantially enough at the molecular scale to affect the structure and function of Mga2. We then determined the FRET efficiency for the K983[D] + K969[A] FRET pair, which reports on the average proximity between the binding site of the E3 ubiquitin ligase Rsp5 (K969[A]) and a target site of ubiquitylation (K983[D]) in the opposing protomer of Mga2. The FRET efficiency in a bilayer with 40 mol% PE was moderately higher than in a PE-free bilayer with an otherwise identical acyl chain composition (50 mol% DOPC, 50 mol% POPC) (Fig. 4d and Supplementary Fig. 4g, h). Expectedly, the inclusion of only 20 mol% PE in the bilayer while maintaining the acyl chain composition affected the FRET efficiency even less (Supplementary Fig. 4h). The highest FRET efficiency was observed for the more saturated membrane (POPC) (Fig. 4d), even though it is less tightly packed (Fig. 4b) and exhibits a similar membrane fluidity as the PE-containing bilayer (Fig. 4a and Supplementary Fig. 4a, e). Thus, the FRET efficiency of this reporter does not correlate with membrane viscosity.

Finally, the functional relevance was studied by an in vitro ubiquitylation assay using the sense-and-response construct (Fig. 4e, f and Supplementary Fig. 4i, j). The highest degree of ubiquitylation was observed for a POPC membrane environment,

which also has the highest degree of lipid saturation. When the sense-and-response construct was reconstituted in a PE-containing bilayer (Supplementary Fig. 4i), which is less saturated but exhibits similar viscosity (Fig. 4a), we observed significantly less ubiquitylation (Fig. 4e, f and Supplementary Fig. 4i, j). Together, these structural and functional data indicate that a key mediator of the homeoviscous response in baker's yeast does not sense membrane fluidity. Instead, they highlight a particular sensitivity of Mga2 to the degree of lipid saturation.

**Configuration and position of lipid double bonds affect Mga2.** To gain deeper insight into the contribution of the double bond in unsaturated lipid acyl chains to the activation of Mga2, we employed a different set of lipids. We used PC lipids with two unsaturated (18:1) acyl chains differing either in the position (Δ6 or Δ9) or the configuration (Δ9-*cis* or Δ9-*trans*) of the double bond (Supplementary Fig. 5a). Expectedly, we find that the "kink" introduced by a *cis* double bond supports membrane fluidity (Fig. 5a and Supplementary Fig. 5b, c) by lowering both lipid packing (Fig. 5b) and membrane order (Fig. 5b). Importantly, Δ6-*cis* acyl chains render the membrane less fluid than Δ9-*cis* acyl chains (Fig. 5a and Supplementary Fig. 5b, c) with no detectable impact on membrane order as studied by C-Laurdan spectroscopy (Fig. 5b and Supplementary Fig. 5d). In contrast, Δ9-*trans* 18:1 acyl chains render the bilayer substantially less fluid (Fig. 5a and Supplementary Fig. 5b) and allow for a much tighter packing of lipids (Fig. 5b and Supplementary Fig. 3a) consistent with the higher gel-to-fluid melting temperature of the lipid (12 °C for Δ9-*trans* 18:1 PC compared with −18 °C for Δ9-*cis* 18:1 PC)[50]. Again, monitoring the changes in fluidity via DPH anisotropy proved less sensitive: increased anisotropies (indicating a decreased mobility) compared with the other bilayer systems were only observed for Δ9-*trans* 18:1 acyl chains containing PC lipids and at temperatures below 12 °C, presumably due the formation of gel phases (Supplementary Fig. 5e)[50]. As additional control, we also characterized the fluidity and lipid packing of a PC bilayer with 16:1 acyl chains and Δ9-*trans* double bonds (Supplementary Fig. 5c–e). Using these bilayer systems differing by the position and configuration of the double bonds in the lipid acyl chains, we set out their impact on the structure and function of Mga2 in vitro.

First, we studied how the double bond position and configuration affects the structural dynamics of Mga2's TMH region using EPR spectroscopy. Placing the sensor in the tightly packed membrane with Δ9-*trans* 18:1 acyl chains caused a substantial broadening of the continuous wave EPR spectra recorded at −115 °C (Fig. 5c) and increased interspin proximities (Fig. 5d). Membrane environments with either Δ6-*cis* or Δ9-*cis* 18:1 acyl chains caused considerably less spectral broadening (Fig. 5c). This indicates that Δ9-*trans* double bonds in lipid acyl chains—more than Δ9-*cis* and Δ6-*cis* bonds—stabilize a rotational orientation of Mga2's TMH region, where spin labels at the position W1042 face each other in the dimer interface[25]. Moreover, our data suggest that lipid acyl chains with Δ9-*trans* double bonds, which are less kinked than those with Δ9-*cis* double bonds, have a similar impact on the structural dynamics of Mga2's TMH as saturated lipid acyl chains.

Next, we tested whether the position and configuration of the double bond in unsaturated lipid acyl chains has an impact on the structural dynamics of Mga2 in the region of ubiquitylation. To this end, we used our FRET reporter (K983[D] + K969[A]) and reconstituted it successfully in different membrane environments as judged by sucrose density gradients (Supplementary Fig. 5f). The reason for the different buoyant densities of the proteoliposomes remains to be fully explored. The FRET signal (Fig. 5e) and

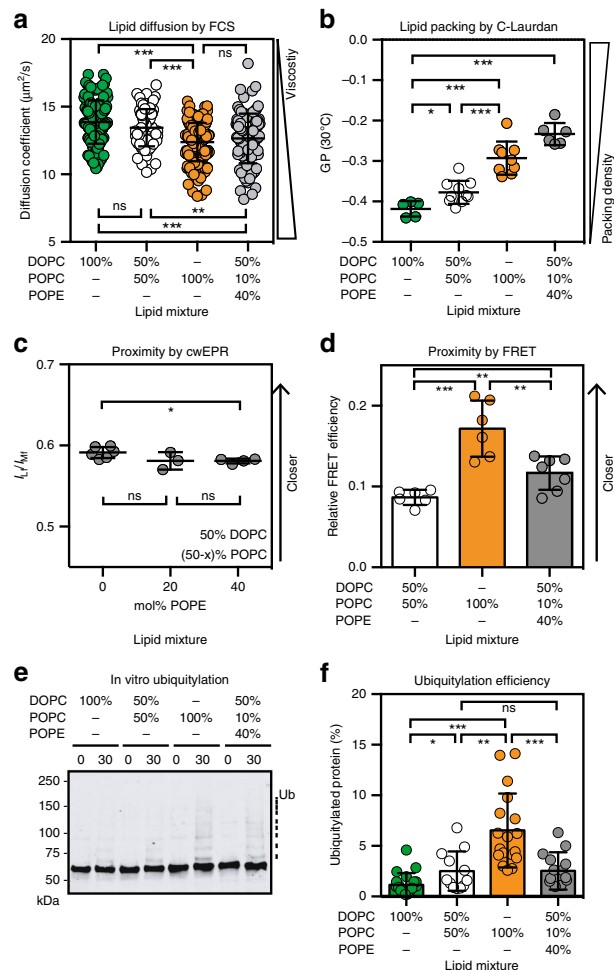

**Fig. 4 The conformation and activity of the sense-and-response construct does not correlate with membrane viscosity. a** Diffusion coefficients of the fluorescent Atto488-DPPE lipid in giant unilaminar vesicles with indicated compositions were determined by confocal point FCS. Data are represented as mean ± SD ($n_{DOPC} = 172$; $n_{(1:1)DOPC:POPC} = 81$; $n_{POPC} = 153$; $n_{40\%POPE} = 100$) and subjected to a Kolmogorov–Smirnov test (*$p < 0.05$, **$p < 0.01$, ***$p < 0.001$). **b** The lipid packing of the indicated lipid compositions were determined by C-Laurdan spectroscopy and expressed as generalized polarization (GP). Data are shown as mean ± SD ($n_{DOPC} = 6$; $n_{(1:1)DOPC:POPC} = 10$; $n_{POPC} = 9$; $n_{40\%PE} = 6$) and analyzed using a two-tailed, unpaired $t$-test (*$p < 0.05$, **$p < 0.01$, ***$p < 0.001$). **c** cwEPR spectra were recorded at −115 °C for a fusion protein composed of MBP and the TMH of Mga2 ($^{MBP}$Mga2$^{1032-1062}$) labeled at position W1042C after reconstitution at a molar protein:lipid ratio of 1:500 in indicated liposomes. The semi-quantitative proximity index $I_{Lf}/I_{Mf}$ indicating the interspin distance was derived from the cwEPR spectra as previously published[25]. Higher values indicate a lower average interspin distance. Data are plotted as mean ± SD ($n_{0\%PE} = 6$; $n_{20\%PE} = 3$; $n_{40\%PE} = 4$) and are analyzed by a two-tailed, unpaired $t$-test (*$p < 0.05$; ns not significant). **d** Relative FRET efficiencies calculated from fluorescence emission spectra (ex: 488 nm, em: 500–700 nm) of the (K983$^D$ + K969$^A$) FRET pair reconstituted in liposomes composed of 50 mol% DOPC, 10 mol% POPC, and 40 mol% POPE. The FRET efficiencies for the other lipid compositions are the same as in Fig. 3g and shown for comparison. Data are plotted as mean ± SD ($n_{(1:1)DOPC,POPC} = 6$; $n_{POPC} = 6$; $n_{40\%PE} = 7$) and analyzed using a two-tailed, unpaired $t$-test (**$p < 0.01$, ***$p < 0.001$). **e** In vitro ubiquitylation of the zipped sense-and-response construct ($^{ZIP-MBP}$Mga2$^{950-1062}$) reconstituted in liposomes of the indicated lipid compositions at a molar protein-to-lipid ratio of 1:8000. After the reaction was stopped, the samples were subjected to SDS-PAGE and analyzed by immunoblotting using anti-MBP antibodies. **f** Densitometric quantification of ubiquitylation from immunoblots as in **e** was performed using Image Studio™ Lite. Data are plotted as mean ± SD ($n_{DOPC} = 20$; $n_{(1:1)DOPC,POPC} = 12$; $n_{POPC} = 18$; $n_{40\%PE} = 14$) and analyzed using a two-tailed, unpaired $t$-test (*$p < 0.05$, **$p < 0.01$, ***$p < 0.001$). Source data are provided as a Source Data file.

FRET efficiency (Fig. 5f) of the reporter were low when the sensor was situated in a bilayer with poorly packing Δ9-*cis* acyl chains (Figs. 3f, g and 5e, f). This indicates a relatively large distance between the binding site for Rsp5 (K969) and the target site for ubiquitylation in the opposing protomer of Mga2 (K983). The FRET efficiency was only mildly higher in a membrane environment with Δ6-*cis* acyl chains (Fig. 5f). This suggests that the position of the *cis* double bond has only a modest impact on the average distance between K969$^A$ and K983$^D$ in the FRET reporter. Significantly higher FRET efficiencies were observed when the reporter was placed in membranes with tightly packing Δ9-*trans* 18:1 acyl chains (Fig. 5f) or Δ9-*trans* 16:1 acyl chains (Supplementary Fig. 5g, h). These findings demonstrate that the structural dynamics of Mga2 is affected by the configuration and potentially by the position of double bonds in unsaturated lipids. Furthermore, our observations on the Δ9-*trans* 16:1 acyl chains indicate a lack of correlation between FRET efficiency (Supplementary Fig. 5g, h), membrane fluidity (Supplementary Fig. 5c), and the lipid-packing density (Supplementary Fig. 5d). Nevertheless, our data suggest a structural coupling and transfer of information from the TMH of Mga2 (Fig. 5c, d) to the site of ubiquitylation (Fig. 5e, f).

To characterize the functional consequences of the proposed membrane-controlled structural dynamics of Mga2, we performed in vitro ubiquitylation assays with the sense-and-response construct ($^{ZIP-MBP}$Mga2$^{950-1062}$) reconstituted initially in three distinct membrane environments (Fig. 5g, h). Although we barely detected any ubiquitylation above background when the sense-and-response construct was reconstituted in a loosely packed bilayer with 18:1 Δ9-*cis* acyl chains (DOPC), we observed a robust ubiquitylation when the construct was situated in a bilayer with either Δ9-*trans* or Δ6-*cis* acyl chains. The highest degree of ubiquitylation of the reporter was observed in the membrane with Δ6-*cis* lipid acyl chains, followed by the less fluid and more tightly packed membrane with Δ9-*trans* lipid acyl chains. This observation supports our previous conclusion that the ubiquitylation of Mga2 does not correlate with membrane viscosity. Furthermore, we find that the FRET efficiencies of the K969$^A$ and K983$^D$ pair in different bilayers (Fig. 5e, f and Supplementary Fig. 5g, h) do not correlate with the respective ubiquitylation efficiencies (Fig. 5g, h and Supplementary Fig. 5i). This is not entirely surprising, because a single FRET pair cannot describe the entire structural dynamics in the region of ubiquitylation. Although increased FRET efficiencies imply an increased average proximity of the two fluorophores, it does not imply a perfect distance and relative orientation of the target lysine residues K$^{980}$, K$^{983}$, and K$^{985}$, and the E3 ubiquitin ligase Rsp5.

Together, our data provide strong evidence that Mga2 does not sense the mere presence or absence of double bonds in the lipid acyl chains. Instead, it is highly sensitive to the configuration and position of the double bond with immediate effect on the structural and dynamic properties of the TMH of Mga2 that dictate the ubiquitylation reaction.

**Bulkiness of sensor residue determines signaling output.** The TMH of Mga2 contains a bulky tryptophan (W1042), which is

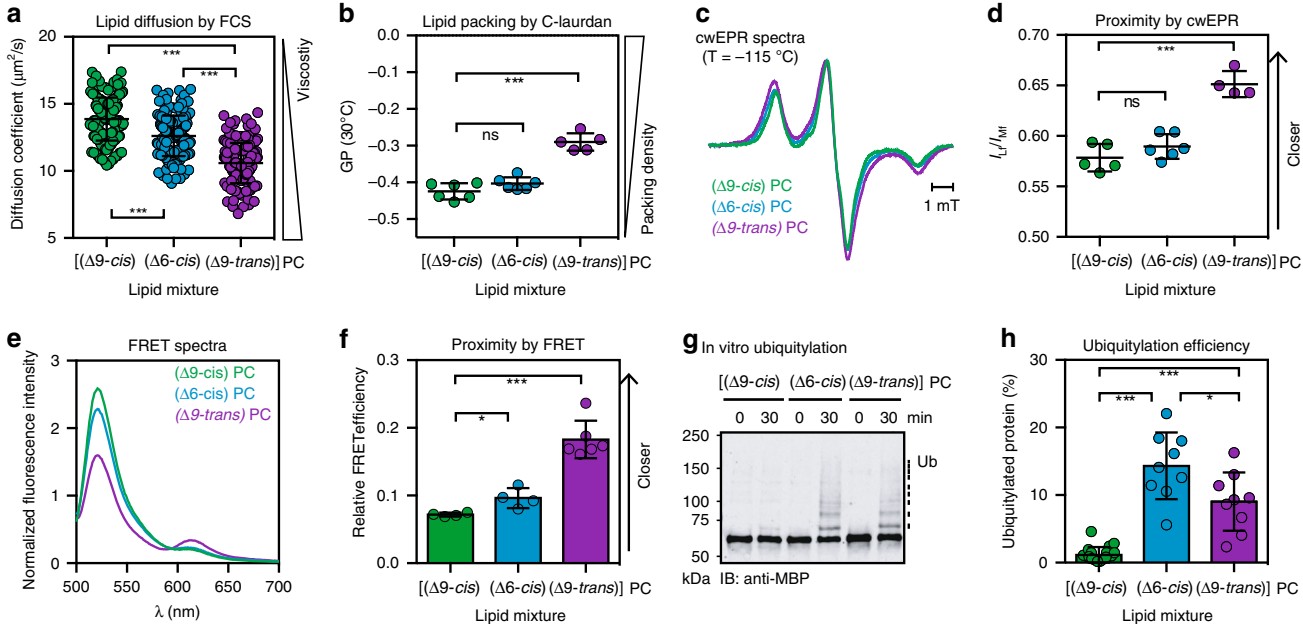

**Fig. 5 The double bond in unsaturated lipid acyl chains affects the configuration and activity of the sense-and-response construct. a** Diffusion coefficients of the fluorescent lipid Atto488-DPPE in GUVs with indicated compositions were determined by confocal point FCS. Data for DOPC (Δ9-*cis*) are the same as in Fig. 4a. Data are plotted as mean ± SD ($n_{(Δ9-cis)} = 172$; $n_{(Δ6-cis)} = 162$; $n_{(Δ9-trans)} = 163$) and analyzed using Kolmogorov–Smirnov tests (***$p < 0.001$). **b** Lipid packing in liposomes was determined by C-Laurdan spectroscopy. Data for DOPC (Δ9-*cis*) are the same as in Fig. 4b. Data are plotted as mean ± SD ($n_{(Δ9-cis)} = 6$, $n_{(Δ6-cis)} = 6$; $n_{(Δ9-trans)} = 5$) and analyzed via unpaired two-tailed, Student's *t*-tests (***$p < 0.001$). **c** Intensity-normalized cwEPR spectra recorded at −115 °C for $^{MBP}Mga2^{1032-1062}$ labeled at position W1042C after reconstitution at a molar protein:lipid ratio of 1:500 in liposomes with indicated composition. **d** Semi-quantitative proximity index $I_{Lf}/I_{Mf}$ derived from cwEPR spectra. High values indicate low average interspin distances. Data are plotted as mean ± SD ($n_{(Δ9-cis)} = 5$; $n_{(Δ6-cis)} = 6$; $n_{(Δ9-trans)} = 4$) and analyzed using two-tailed, unpaired *t*-tests (***$p < 0.001$). **e** Fluorescence emission spectra of the membrane-reconstituted FRET pair (K983$^D$ + K969$^A$) (ex: 488 nm, em: 500–700 nm) are plotted after normalization to the maximal emission upon acceptor excitation (ex: 590 nm). Data for DOPC (Δ9-*cis* PC) are the same as in Fig. 3f. **f** Relative FRET efficiencies were calculated as in **e**, plotted as mean ± SD ($n_{(Δ9-cis)} = 4$; $n_{(Δ6-cis)} = 4$; $n_{(Δ9-trans)} = 6$), and analyzed by two-tailed, unpaired *t*-tests (*$p < 0.05$; **$p < 0.005$). Data for DOPC (Δ9-*cis* PC) are the same as in Fig. 3g. **g** In vitro ubiquitylation of the zipped sense-and-response construct ($^{ZIP-MBP}Mga2^{950-1062}$) reconstituted in indicated liposomes at a molar protein-to-lipid ratio of 1:8000 and at 30 °C. After stopping the reaction, the samples were subjected to SDS-PAGE and analyzed by immunoblotting using anti-MBP antibodies. **h** Densiometric quantification of ubiquitylation at the indicated time points from immunoblots as in **g**). Data are plotted as mean ± SD ($n_{(Δ9-cis)} = 20$; $n_{(Δ6-cis)} = 9$; $n_{(Δ9-trans)} = 9$. Data for DOPC (Δ9-*cis* PC) are the same as in Fig. 4f. Source data are provided as a Source Data file.

functionally important and might serve as a sensor residue[25]. Previous MD simulations have shown that W1042 is situated in the hydrophobic core of the bilayer overlapping with the Δ9-*cis* double bonds of unsaturated phospholipids[25]. We hypothesized that Mga2 senses the lipid-packing density in this region of the lipid bilayer and/or a thin slice of the lateral compressibility profile[2,18]. The sensitivity of our sense-and-response construct to the position and configuration of the double bond in lipid acyl chains (Fig. 5) is consistent with this idea. Our model predicts that the activation of Mga2 is controlled by the size of the amino acid side chain at position 1042, which also controls the population of alternative, rotational configurations of the sensory TMH in a dynamic equilibrium. An increased packing and a lowered lateral compressibility in this region should cause a relatively large amino acid to "hide" in the dimer interface, thereby stabilizing a productive configuration. A smaller residue should be less sensitive to the membrane environment and populate non-productive configurations.

To test this prediction, we have substituted W1042 with either tyrosine (Y), phenylalanine (F), glutamine (Q), leucine (L), or alanine (A), and assayed the role of the side-chain bulkiness and aromatic character on the signal output in vivo. Expectedly, a Δ*SPT23*Δ*MGA2* double mutant lacking both transcriptional regulators of *OLE1* does not grow unless UFAs were provided with the medium (Fig. 6a)[51]. This UFA auxotrophy of

Δ*SPT23*Δ*MGA2* cells is complemented by both wild-type and mutant *MGA2* variants expressed from the endogenous promotor on a *CEN*-based plasmid (Fig. 6a, b). However, the growth of these cells was highly dependent on the amino acid at position 1042 under UFA-limiting conditions (Fig. 6a and Supplementary Fig. 6a, b). We observed a striking correlation between the size of the side chain and the optical density of overnight cultures (Fig. 6b). The only exception to this near-perfect correlation was the W1042Q mutation. Given that intra-membrane glutamines are known to mediate homotypic interactions[52], we speculate that the W1042Q mutation stabilizes a rotational conformation of the TMHs, where the two Q1042 side chains face each other and interact, thereby stabilizing Mga2 in a productive configuration. Furthermore, the phenotypic differences between the W1042Q, W1042L, and W1042A variants (Fig. 6a–d) show that an aromatic character at the sensory position is not absolutely required for sensing.

Next, we studied the impact of these mutations on the proteolytic processing of full-length Mga2 in cells (Fig. 6c, d). We found a perfect concordance of these immunoblot experiments with the in vivo phenotypes (Fig. 6a, b). The processing of the membrane-bound precursor of Mga2 (P120) to the signaling-active form (P90) was greatly affected by the residue at the position 1042. These data were complemented by functional in vitro experiments using the sense-and-response construct

reconstituted in liposomes (Supplementary Fig. 6c, d). The in vitro ubiquitylation in a POPC bilayer was significantly reduced to almost background levels by the W1042A mutation. These data support a central, sensory role of W1042.

Using a previously established MD simulation pipeline[25], we analyzed the impact of the W1042Y, W1042Q, W1042F, and W1042A mutations on the structural dynamics of dimeric TMHs derived from Mga2. We performed extensive, coarse-grained MD simulations of these mutant variants in a POPC bilayer (Supplementary Fig. 6e–h), which we have shown to support robust ubiquitylation of Mga2 in vitro (Fig. 2e, f). At least three populations of distinct configurations in the TMH region were detected. They correspond to configurations with the two residues at the position 1042 (i) facing each other, (ii) pointing in the same direction, or (iii) pointing away from each other. The population of these configurations were clearly affected by mutations at the position of W1042 (Supplementary Fig. 6f–i). Notably, the mutants with a particularly strong impact on the structural dynamics in the TMH region (W1042F and W1042A) also exhibit a strong functional defect in vivo (Fig. 6a–d and Supplementary Fig. 6e–h). Based on these data, we conclude that W1042 acts as a sensor residue and that the size and the chemical character of the amino acid in this position controls the signaling output.

**Representing local lipid packing reveals sensing mechanism**. As membrane fluidity is unlikely to control the activation of Mga2, and as the size of the residue 1042 determines the functionally relevant structural dynamics in the transmembrane region, we turned our attention to the local packing density of the lipids.

To explore the contribution of lipid headgroups and the lipid acyl chain composition on the investigated bilayer systems, we performed extensive all-atom MD simulations of protein-free bilayers. As a reference, we first characterized a DOPC bilayer containing no saturated lipid acyl chains (Supplementary Fig. 3a–d). This bilayer was compared with three other systems: a 50:50 mixture of DOPC and POPC containing 25% saturated lipid acyl chains, POPC with 50% saturated lipid acyl chains, and a 50:10:40 mixture of DOPC, POPC, and POPE, which contain 25% saturated acyl chains, but almost half of the lipids have a smaller ethanolamine headgroup (Supplementary Fig. 3a–d). We asked whether an increased lipid packing due to changes in the acyl chain composition is qualitatively distinct from increased lipid packing due to different lipid headgroup compositions. Of particular interest is a region between 3 Å and 10 Å from the bilayer center, where the sensory W1042 is situated as previously shown[25]. We determined the 3-dimensional number density of lipid atoms in small $1 Å \times 1 Å \times 1 Å$ voxels (cubes) within the DOPC bilayer as a measure for the local packing density (Fig. 7a) and did the same for the other three bilayer systems (Supplementary Fig. 7). Expectedly, this representation identifies a particularly high density of lipid atoms in the headgroup region and a much lower one in the bilayer center.

As we could not detect any ubiquitylation of the sense-and-response construct in DOPC, but robust ubiquitylation in POPC (Fig. 4e, f), we turned our attention toward the differences between DOPC and the other bilayer systems. We derived difference maps of the local packing density between DOPC and the other three bilayer systems to highlight regions of increased and decreased local number densities (Fig. 7b). We reasoned that any difference in the local packing density, especially in the region of the sensory W1042, might contribute to regulation of Mga2. We find that saturated lipid acyl chains increase the density of lipid atoms relative to DOPC precisely in this region of the bilayer (3 Å–10 Å from the bilayer center) (Fig. 7b). In a bilayer containing 40 mol% PE, we also observe

increased densities along the acyl chain region, but this increase is less pronounced and more delocalized (Fig. 7b). This computational analysis demonstrates that saturated lipid acyl chains increase the local density of lipid atoms precisely and specifically in the region of the sensory tryptophan W1042. Based on these findings and our experimental data, we propose that Mga2 senses a bilayer property at the level of the bulky W1042, which is closely related to the local packing density.

**Discussion**
We have reconstituted key steps of sensing and communicating lipid saturation by the prototypical type II membrane property sensor Mga2[19]. We uncover a unique sensitivity of Mga2 for the lipid acyl chain composition of the ER membrane and provide direct evidence for a functional coupling between the dimeric, sensory TMHs and the sites of ubiquitylation. Our in vitro system allowed us to directly test a central assumption underlying the concept of homeoviscous adaptation[9,15,17]. By investigating the role of membrane fluidity on the ubiquitylation of Mga2, we demonstrate that the core regulator of fatty acid desaturation in baker's yeast[21,53] is not regulated by the viscosity of the membrane (Figs. 4, and 5). Instead, our data suggest that Mga2 uses a bulky TMH residue (W1042) to sense the packing density, which probably affects the lateral compressibility profile at a specific depth in the ER membrane. Based on our findings, we conclude that membrane fluidity does not serve as the central measured property for regulating the lipid acyl chain composition in baker's yeast and presumably many other eukaryotic species.

Our in vitro approach with reconstituted proteoliposomes has provided insights into the sensitivity of Mga2 to physiologically relevant changes of the lipid acyl chain composition[8,44]. Baker's yeast can synthesize only Δ9-cis mono-UFAs and the average proportion of these unsaturated acyl chains lies between 65% and 75% in glycerophospholipids depending on the temperature of cultivation[8]. The sense-and-response construct cannot be ubiquitylated in a relatively unsaturated membrane (75% Δ9-cis 18:1 acyl chains), but it is robustly ubiquitylated in POPC, which is slightly more saturated (50% Δ9-cis 18:1 acyl chains) (Fig. 2f). A simple back-of-an-envelope calculation that considers only the volume of the lipid bilayer highlights the remarkable dose-response relationship of this machinery: the sense-and-response system is OFF, when the concentration of unsaturated lipid acyl chains is ~1.9 M, but it is ON at a concentration of ~1.3 M (assumptions: ~370,000 lipids per 200 nm liposome; ~4,82 $\times 10^{-19}$ l membrane volume per liposome). This switch-like response is based on fluctuating signals from the membrane, which are decoded by the sensor protein into an almost binary output.

To better understand the nature of sensing, we have employed a strategy to compare different bilayer systems by means of the local number density of lipid atoms (Fig. 7 and Supplementary Fig. 7). This approach yielded insight into the qualitatively distinct impact of saturated lipids and a perturbed PC-to-PE ratio on the lipid-packing densities in the core of the membrane (Fig. 7b). We find that increased lipid saturation causes a localized, sharp increase of the local packing density, whereas the inclusion of PE into the bilayer causes a rather mild increase of the packing density, which is delocalized along the lipid acyl chains (Fig. 7b and Supplementary Fig. 7). It is well known that the inclusion of PE into a PC-based bilayer causes intrinsic curvature stress and characteristic changes of the lateral pressure profile (Supplementary Fig. 3c), which may be sensed particularly well by hourglass-shaped transmembrane domains[54–57]. However, the cylindrical TMHs of Mga2 dimers are more sensitive to saturated lipid acyl chains causing an increased packing density in a specific region of the bilayer overlapping with the sensory W1042 residue

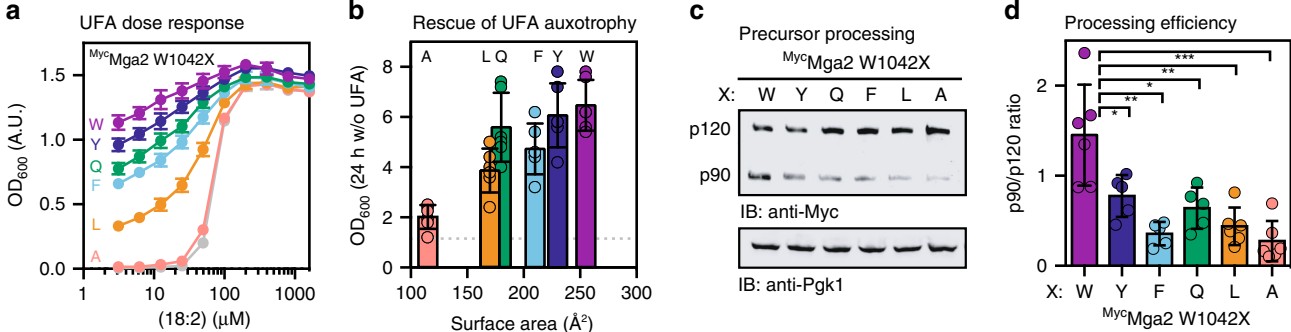

**Fig. 6 The activity of Mga2 is tuned by mutations in the sensory TMH. a** Dose-dependent rescue of UFA auxotrophy by linoleic acid (18:2). $\Delta SPT23\Delta MGA2$ strains carrying $CEN$-based plasmids to produce $^{Myc}$Mga2 variants with the indicated residues at position 1042 were cultivated for 16 h at 30 °C in SCD-Ura medium supplemented with indicated concentrations of linoleic acid in 0.8% tergitol. The density of the culture was determined at 600 nm ($OD_{600}$) and plotted against the concentration of linoleic acid. Cells carrying an empty vector served as control (gray). Plotted is the mean ± SEM ($n = 8$). **b** Rescue of UFA auxotrophy of $\Delta SPT23\Delta MGA2$ by Mga2 variants. Cells producing mutant Mga2 as in **a** were cultivated for 24 h in the absence of supplemented UFAs in SCD-Ura medium. Cell density was determined as in **a** and was plotted against residue surface area of residues installed at position 1042[66]. Plotted is the mean ± SEM of five independent experiments. The dotted line indicates the OD measured for an empty vector control. **c** Immunoblot analysis of the Mga2 processing efficiency. Wild-type cells (BY4741) producing the indicated $^{Myc}$Mga2 variants at position 1042 were cultivated in full medium (yeast extract–peptone–dextrose; YPD) to the mid-exponential phase. Cell lysates were subjected to SDS-PAGE and analyzed via immunoblotting using anti-Myc antibodies to detect the unprocessed (p120) and the processed, active form (p90) of Mga2. An immunoblot using anti-Pgk1 antibodies served as loading control. **d** Densitometric quantification of the ratio of p90:p120 in immunoblots as in **c**. Signal intensites were quantified using Fiji and plotted as mean ± SD ($n_W = 6$, $n_Y = 5$, $n_F = 5$, $n_Q = 5$, $n_L = 6$, $n_A = 6$)[67]. A two-tailed, unpaired $t$-test was performed to test for statistical significance (*$p < 0.05$, **$p < 0.01$, ***$p < 0.001$). Source data are provided as a Source Data file.

(Fig. 7a, b). We thus provide not only a model for lipid sensing by Mga2 but also an approach to dissect the relative impact of lipid headgroup and the acyl chain composition on membrane protein function in general.

Our results lead to the following model of lipid saturation sensing: the lipid acyl chain composition has a profound impact on the distribution of lipid atoms in the hydrophobic core of the membrane (Fig. 7a, b and Supplementary Fig. 7). Mga2 senses increased packing densities via a bulky tryptophan residue (W1042) situated deep in the hydrophobic core of the bilayer (3 Å–10 Å from the bilayer center[25]). In a more saturated membrane, the packing density increases at the level of the sensory tryptophan (W1042) (Fig. 7b), which then rotates away and "hides" in the dimer interface (Fig. 1a). In a more unsaturated membrane, the lower packing density in this region allows the sensory tryptophan to point on average more often towards the lipid environment[25] (Fig. 1a). We would like to stress that the TMHs of Mga2 constantly rotate against each other and explore a large variety of rotational states (Supplementary Fig. 6e–h) in each membrane but the specific lipid composition of the membrane has a significant impact on the structural dynamics in the TMH region by determining the relative probability of these rotational states[25]. It seems that the fluctuating signals from the membrane are decoded by the structural dynamics of the TMHs and then transmitted to the site of ubiquitylation via a disordered, juxtamembrane region (Figs. 1b and 3g). We speculate that the flexible linkage provides a means to bias the orientation and relative position of two "ubiquitylation zones" around the E3 ubiquitin ligase Rsp5 bound to Mga2, however, with a minimal perturbation of the TMH dynamics. Such "zones of ubiquitylation" have recently been predicted for Rsp5 and implicated into the quality control of misfolded and mistargeted plasma membrane proteins[58]. Supported by our FRET data (Fig. 3f, g), we propose that Rsp5 bound to one protomer of dimeric Mga2 can ubiquitylate specific lysine residues on the other protomer, when it is properly placed and oriented. Effectively, this *trans*-ubiquitylation would be controlled by the physicochemical properties of the ER membrane. The remarkable sensitivity of

Mga2 ubiquitylation to the lipid environment might be sharpened by deubiquitylating enzymes[59] such as Ubp2[60] and supported by an activating, *trans*-autoubiquitylation of Rsp5[61].

The assays and tools established here provide handles to better understand the structural and dynamic features that render a protein a good substrate of the E3 ubiquitin ligase Rsp5. Identifying the molecular rules of substrate selection is a major open question, because Rsp5 has been implicated in most diverse aspects of cellular physiology including endocytosis[58], mitochondrial fusion[62], and the turnover of heat-damaged proteins in the cytosol[63]. Our in vitro system using a membrane-reconstituted, conditional substrate of Rsp5 provides a unique opportunity to better understand (i) the contribution of *trans*-autoubiquitylation of Rsp5, (ii) the relevance of structural malleability in Rsp5 substrates, and (iii) the role of deubiquitylating enzymes in defining the selectivity and sensitivity of the Rsp5-mediated ubiquitylation. In the context of the Mga2 sensor, an intriguing question is how "noisy" signals from the TMH region are transduced into robust, almost switch-like ubiquitylation responses.

Two lines of evidence suggest that the rotation-based sensing mechanism of Mga2[25] is based on a collective, physical property of the membrane rather than on a preferential, chemical interaction with the double bonds in the lipid acyl chains. First, Mga2 distinguishes robustly between two membrane environments that differ in the configuration of the double bonds (*cis* or *trans*) in the lipid acyl chains, but not in the overall abundance of double bonds (Fig. 5). Second, an aromatic amino acid, which might confer some chemical specificity for double bonds, is not absolutely required at the position of the sensory tryptophan (W1042) in the TMH (Fig. 6). A partial activity of the OLE pathway is preserved when the sensor residue W1042 is substituted with leucine, but not when it is substituted with the smaller alanine (Fig. 6a, b; and Supplementary Fig. 5b). Nevertheless, our data do not rule out an important contribution of chemical specificity to the sensor function. We expect that the high degree of structural malleability in the TMH region and at the site of ubiquitylation is established by a fine balance of chemical interactions and collective, physical membrane properties. In fact, even the rather

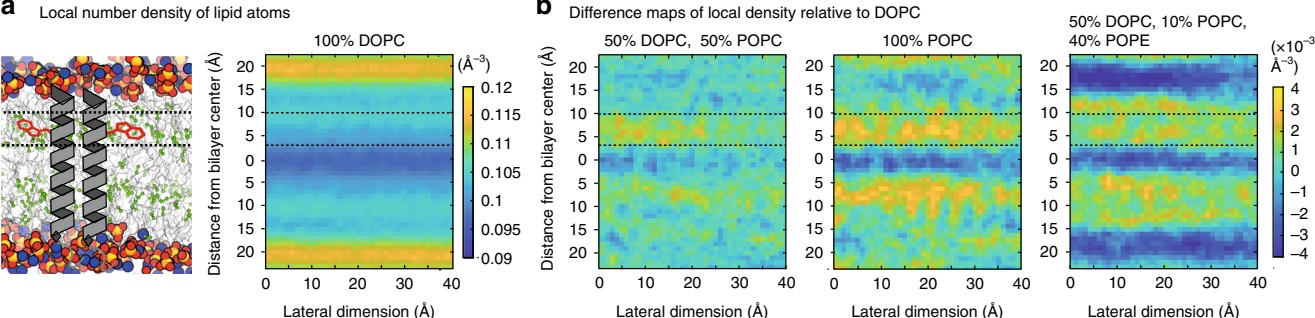

**Fig. 7 The local number density of lipid atoms in a specific region of the membrane correlates with Mga2 activation. a** Snapshot from an all-atom MD simulation of a protein-free DOPC bilayer (left) and the derived local number density of lipid atoms (right). A schematic representation of the dimeric TMH of Mga2 in the left panel is provided to indicate the relative position of the sensory tryptophan W1042 (red sticks and dashed line). The position of double bonds in the lipid acyl chains is given by green spheres, whereas a schematic representation of the Mga2-TMH is given to guide the eye. The number density of lipid atoms in cubic boxes with a side of 1 Å was calculated and plotted (right panel) for different depths in the bilayer (distance for bilayer center) and different lateral positions (lateral dimension). Highest local number densities are indicated in yellow and are observed in the region of the lipid headgroups. Lowest densities are indicated in dark blue and observed in the center of the lipid bilayer. For details see the Supplementary Materials. **b** Difference maps of the local density were determined by subtracting the local number density of lipid atoms of DOPC from the one of the indicated bilayer systems. Increased packing densities relative to the DOPC bilayer are indicated in yellow, whereas decreased densities are indicated in blue. The region probed by the sensory tryptophan[25] is indicated by two dashed lines. Changes in lipid saturation and the lipid headgroup region have qualitatively distinct impact on the distribution of local packing densities. Source data are provided as a Source Data file.

conservative mutation W1042F has a substantial impact on the structural flexibility in the TMH region (Supplementary Fig. 6e) and on the functionality of the OLE pathway in vivo (Fig. 6a–d).

In conclusion, we have provided detailed mechanistic insight into a sensory system that is centrally important for membrane adaptivity. Our findings challenge the common view of membrane fluidity as pivotal measured variable in eukaryotic cells and have important implications to all processes involving membrane lipid adaptation. Beyond that, our work represents an important step towards identifying the molecular rules of substrate selection by the E3 ubiquitin ligase Rsp5. This work opens a door towards establishing genetically encoded machineries that can sense specific membrane features, which are indiscernible by conventional tools. In the future, these sensors will be exploited to dissect the physical membrane properties of different organelles and cells in vivo and in real-time.

## Methods

**Plasmids and strains and oligonucleotides**. All plasmids and strains used in this study are listed in Supplementary Tables 1 and 2. Oligonucleotides used for molecular cloning are listed in Supplementary Table 3.

**Antibodies**. Antibodies used for immunoblotting are listed in Supplementary Table 4.

**Expression and purification and labeling of ^MBPMga2 fusions**. The ^ZIP-^MBPMga2^950–1062 fusion protein comprising the leucine zipper of the Gcn4 transcription factor (residues 249–281), the MBP from *Escherichia coli*, and the residues G950–D1062 from Mga2 was generated using the pMal-C2x plasmid system. The resulting constructs were produced in *E. coli* and isolated in detergent solution using amylose affinity followed by a preparative SEC (Superdex 200 10/300 Increase). For fluorescent labeling, the K983C and K969C variants were incubated with 1 mM ATTO488 or ATTO590 (ATTO-TEC GmbH) on the affinity purification column for 16 h at 4 °C. The ^MBPMga2^1032–1062 fusion protein containing residue R1032-D1062 from Mga2 and a W1042C mutation was purified and labeled with MTS spin probes. For details, see Supplementary Materials. The proteins were stored in 40 mM HEPES (pH 7.0), 120 mM NaCl, 0.8 mM EDTA, 40 mM OG, and 20% (w/v) glycerol.

**Reconstitution of ^MBPMga2 fusions in proteoliposomes**. The spin-labeled ^MBPMga2-TMH fusion was reconstituted at a protein:lipid molar ratio of 1:500. For details see Supplementary Materials. The unlabeled or ATTO-labeled ^ZIP-^MBPMga2^950–1062 constructs were reconstituted at different protein–lipid molar ratios of 1 to 5000, 1 to 8000, and 1 to 15,000. To this end, lipids (final

concentration 1 mM) and OG (final concentration 37.5 mM) were mixed with either labeled or unlabeld proteins in a final volume of 1 ml. After 10 min of incubation at room temperature under constant agitation, the detergent was via SM-2 biobeads (two-step removal using 500 mg and 100 mg, respectively). The yield of the reconstitution was tested after harvesting the proteoliposomes and it was comparable for all the desired protein and lipid compositions used in this study. The average diameter of the proteoliposomes was determined at 25 °C by dynamic light scattering (Malvern Zetasizer Nano S90) in 20 mM Hepes, pH 7.4, 145 mM NaCl, 5 mM MgCl₂, 5% (w/v) glycerol. A detailed description for the reconstitution is provided in the Supplementary Materials.

**Diffusion coefficients by FCS**. FCS on the GUVs was carried out using Zeiss LSM 880 microscope, ×40 water immersion objective (numerical aperture 1.2)[47]. First, GUVs were labeled by adding fluorescent analogs to a final concentration of 10–50 ng/mL (≈0.01 mol%). To measure the diffusion on the GUV membrane, vesicles were placed into an eight-well glass-bottom (#1.5) Ibidi chambers coated with bovin serum albumin. GUVs of small sizes (≈10 μm) were picked for measurements. The laser spot was focused on the top membrane of the vesicles by maximizing the fluorescence intensity. Then, three to five curves were obtained for each spot (5 s each). The obtained curves were fit using the freely available FoCuS-point software using two-dimensional and triplet model[64].

**C-Laurdan spectroscopy**. C-Laurdan was used to measure lipid packing[49]. To this end, 333.3 μM lipid was mixed with 0.4 μM C-Laurdan dye in 150 μl 50 mM HEPES pH 7.4, 150 mM NaCl, 5 %(w/v) glycerol. The sample was excited at 375 nm and an emission spectrum from 400 to 600 nm was recorded (excitation and emission bandwidth 3 nm). For blank correction, an emission spectrum recorded in the absence of C-Laurdan was used. The GP, which ranges theoretically from +1 for most ordered to −1 for most disordered membranes, was calculated by integrating the intensities between 400 and 460 nm ($I_{Ch1}$), and 470 and 530 nm ($I_{Ch2}$), and according to (1).

$$GP = \frac{I_{Ch1} - I_{Ch2}}{I_{Ch1} + I_{Ch2}} \qquad (1)$$

**Recording and analysis of FRET spectra**. For FRET measurements, the ^ZIP-^MBPMga2^950–1062 K983^ATTO488 and ^ZIP-MBPMga2^950–1062 K969^ATTO590 constructs were used as fluorescence donor and acceptor, respectively. Fluorescence emission spectra were recorded in detergent solution and in proteoliposomes at 30 °C. The samples were excited at 488 nm and 590 nm for donor and acceptor excitation, respectively. The spectra were normalized to the maximal acceptor fluorescence intensity after direct excitation to correct for subtle variations in the reconstitution yields. As the bleed-through for both the donor and acceptor fluorescence was negligible, ratiometric FRET (relative FRET: $E_{rel}$) was determined as the donor-to-acceptor intensity ratio at 525 nm and 614 nm from the raw data (2) for qualitative

comparisons.

$$E_{rel} = \frac{I_A}{I_D + I_A} \qquad (2)$$

**In vitro ubiquitylation assay**. Proteoliposomes containing $^{ZIP-MBP}Mga2^{950-1062}$, $^{8xHis}$Ubiquitin (see Supplementary Materials for a description of expression and purification), cytosol, and an 10× ATP-regenerating system were mixed on ice in a total volume of 20 μl, to obtain final concentrations of 0.1 μM $^{ZIP-MBP}Mga2^{950-1062}$, 0.1 μg/μl $^{8xHis}$Ubiquitin, 1 μg/μl cytosolic proteins, 1 mM ATP, 50 mM creatine phosphate, and 0.2 mg/ml creatine phosphokinase in ubiquitylation buffer (20 mM HEPES pH 7.4, 145 mM NaCl, 5 mM MgCl₂, 10 μg/ml chymostatin, 10 μg/ml antipain, 10 μg/ml pepstatin). Cytosol was prepared from BY4741 cells grown to mid-log phase ($OD_{600} = 1$) in YPD medium[42]. For details see Supplementary Materials. The ubiquitylation reaction was incubated at 30 °C and stopped by mixing the sample at a ratio of 2:1 with 5× reducing sample buffer (8 M urea, 0.1 M Tris-HCl pH 6.8, 5 mM EDTA, 3.2% (w/v) SDS, 0.15% (w/v) bromphenol blue, 4% (v/v) glycerol, 4% (v/v) β-mercaptoethanol) and boiling it. Protein ubiquitylation was analyzed by SDS-polyacrylamide gel electrophoresis using 4–15% Mini-PROTEAN-TGX gels (BioRad) and immunoblotting using anti-MBP antibodies. Uncropped and unprocessed scans of immunoblots are provided as Source Data file and Supplementary Source Data file.

**Molecular dynamics simulations**. We characterized the impact of mutations on the assembly of the dimer-forming TMH of Mga2 by performing coarse-grained MD simulations. We modeled the TMHs containing the mutations W1042F, W1042Y, and W1042Q, and simulated their dynamics in a POPC bilayer for 1 ms, 0.93 ms, and 1 ms, respectively[25]. For details, see Supplementary Materials.

Furthermore, we examined the biophysical properties (lipid packing, acyl chain order, and local density) of the experimentally studied lipid bilayers with all-atom MD simulations. The simulations were constructed, simulated, and analyzed mostly based on existing protocols[56]. For details, see Supplementary Materials. Details on system size, trajectory length, and analysis can be found in Supplementary Table 5.

**Reporting summary**. Further information on research design is available in the Nature Research Reporting Summary linked to this article.

## Data availability
Data supporting the findings of this manuscript are available from the corresponding author upon reasonable request. A reporting summary for this Article is available as a Supplementary Information file.

The source data underlying Figs. 1b, 2b–f, 3b–h, 4a–f, 5a–h, 6a–d and Supplementary Figs. 1a–e, 2a–d, 3a–d, 4a–j, 5b–i, 6b–h provided as a Source Data file.

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

## Acknowledgements

We thank Laura Glück and Kim Wendrich for excellent technical assistance. We acknowledge Jeffrey Brodsky and Volker Dötsch for sharing reagents and protocols, as well as Edward Lyman and Alex Sodt for fruitful discussions. This work was supported by the Deutsche Forschungsgemeinschaft (DFG, ER608/2−1) to R.E. and (HA6322/3−1) to I.H., the Volkswagen Foundation (Life?, grant numbers 93089 and 93091) to R.E. and I.L., respectively, and the European Molecular Biology Organization (EMBO, ASTF 451−2014) to S.B. E.S. is funded by British Council Newton-Katip Celebi Fund (#352333122). I.L. and M.D. acknowledge funding from the NIH/National Institute of General Medical Sciences (GM114282, GM124072, and GM120351) and the Human Frontiers Science Program (RGP0059/2019). R.C. and G.H. were supported by the Max Planck Society. This work used the Extreme Science and Engineering Discovery Environment (allocation TG-MCB180168), which is supported by National Science Foundation grant number ACI-1548562. Data regarding the wild-type and the W1042A variant of the Mga2-TMH in Supplementary Fig. 5 are replotted from Covino et al.[25], with permission from Elsevier.

## Author contributions

Conceptualization: R.E. Experimental design: S.B., E.S., D.W., I.H., R.C., M.C., J.R., and R.E. Performed experiments: S.B., R.C., M.D., E.S., D.W., and J.R. Writing—original draft: S.B. and R.E. Writing—revised draft: R.E., S.B., R.C., and M.D. Funding acquisition: R.E., I.H., E.S., I.L., and G.H. Supervision: R.E., G.H., and I.L.

## Competing interests

The authors declare no competing interests.
