## [Peer Review File · Nature Communications]

Reviewers' Comments:

Reviewer #1:

Remarks to the Author:

The manuscript by Ballweg and colleagues addresses a very timely and important question in membrane biology, i.e. how is the membrane saturation state sensed. In a previous publication the authors have shown that in yeast the ER-resident membrane protein Mga2 protein acts as a sensor for membrane lipid saturation to maintain a constant degree of membrane fluidity. Based on complementary cellular, in vitro and in silico (MD simulations) studies the authors suggested a transmembrane helix rotation-based mechanism of sensing lipid saturation, with a key role of a tryptophan residue that adopts different orientations depending on the degree of lipid saturation. In essence, a saturated lipid environment stabilizes a dimer conformation with the two tryptophans of each protomer facing towards the dimerization interface, leading to efficient ubiquitylation of the cytosolic protein domain, subsequent proteolytical digest and release of a fragment that in the nucleus activates transcription of the desaturase Ole1.

In the current manuscript the authors now advance these studies to elucidate the molecular mechanism underlying Mga2's sensor activity. To this end the authors have established an elegant in vitro assay to measure ubiquitylation of a so-called sense-and-response reporter containing a truncated version of Mga2. Using this assay the authors investigated the effect of different liposomal membrane properties to decipher biophysical parameter, such as viscosity and lateral pressure profiles, for their ability to modulate ubiquitylation efficiency. In addition, a FRET approach was used to correlate ubiquitylation efficiency with proximity of E3 ubiquitin ligase binding site and ubiquitin acceptor site within the cytosolic domain of Mga2. From their data the authors suggest that Mga2 senses the lateral pressure profile at a specific site within the bilayer but not the overall state of membrane viscosity/fluidity.

This work significantly adds to our understanding of molecular mechanisms contributing to membrane homeostasis.

Comments

1. As acknowledged by the authors it is a major challenge to study different membrane properties separately due to a high degree of interdependency, which raises some questions concerning the interpretation of the results. To support the model proposed by the authors and to get further insight at a molecular level, the work would benefit from a complementary method such as MD simulations.

2. Could the author comment on potential structural requirements to the TM domains to sense changes in lateral pressure profiles – such as hourglass shaped TM structures.

3. Due to the strong interdependency of biophysical parameters dictating membrane properties, I think the number and type of different conditions chosen are not yet entirely convincing to support the authors conclusion that only lateral pressure is sensed by Mga2. At this stage I would rather conclude that both parameters are sensed.

As an example, the authors argue that PE is a factor perturbing membrane viscosity. Increasing amounts of PE were shown not to effect cwERP values, taken as one indicator that viscosity is not parameter sensed by Mga2. However, PE was reported to not only affect membrane viscosity but also the lateral pressure profile. Can the authors rule out a contribution to lateral pressure profiles? In addition to the cwERP data, proximity by FRET, C-Laurdan, lipid diffusion and ubiquitylation data should be shown for the PE titration experiments to support the authors conclusion. Temperature-dependent changes in lipid profiles in *S. cerevisiae* were reported to significantly alter the PE:PC ratio. Can the authors comment on possible roles of lipid headgroups in Mga2's sensor function?

4. The data on amino acid requirements at the sensor site is intriguing. How do these variants behave in the FRET proximity and ubiquitylation assay, especially concerning the F,Y and Q

variants? Do these variants show the same dependency on membrane properties as the wildtype (e.g. comparison of DOPC vs POPC).

5. Lines 333ff, comparing Figure 4 and 5: Can the authors comment on the correlation of proximity by FRET versus ubiquitylation efficiency.

6. Lines 343ff: Can the authors comment on the biological implication, i.e. range of db positions in S.c. membranes? How does delta9-C16:1 cis/trans behave?

7. Are the Mga2 reconstitution efficiencies comparable for the various conditions used to generate proteoliposomes?

8. Line 163: Only a protein:lipid ratio of 1:8,000 is shown in that Figure.

9. Figure 2E: A protein-to-lipid ratio of 1:5,000 is used here but in most other experiments it is 1:8,000. I would suggest to use the same protein-to-lipid ratio in same types of experiments and move comparisons of different ratios into the Supplement.

10. Figure 3G: filling of bar graph could be changed to white to fit with Figure 4 data.

11. Figure 4A: I would suggest to replace the arrow indicating increasing viscosity by a triangle as used in B for describing the extent of packing density.

12. Figure 4B: Check x-axis labeling

13. Figure S4C: The author should comment on the strikingly different properties of liposomes (used for the experiments described in Figure 5) in sucrose gradient centrifugations.

14. Check legends to Figure 4 and 5: data refer to Figure 3F-G not 2F-G.

15. Figure 4-5: The authors describe diffusion coefficients of lipids in the in vitro setting. These data should be discussed in comparison to values obtained in cellular membranes/in presence of transmembrane domain, i.e. how does the range of viscosity tested relate to the in vivo situation.

16. Figure 6E: Why was the protein:lipid ratio changed to 1:15,000 as compared to 1:8,000 as used in the experiments shown in the main text figures?

17. In order to perform the cwERP measurements the authors had to use a protein:lipid ratio of 1:500. How does this affect the protein/read out parameters?

18. Can the authors exclude that the presence or absence of the Leucine zipper does not influence dynamics of the TM helices?

19. Line 396: unique sensitivity...of..

20. Line 1043: check sentence

Reviewer #2:

Remarks to the Author:

Stephanie Ballweg et al. address one of the key questions in membrane biology in their manuscript: How do cells measure the unsaturation degree in their membranes? It has often been assumed that cellular sensors react to changes in membrane viscosity, which is correlated with unsaturation degree, but the actual data basis for this assumption has been remarkably weak. The

authors now make a convincing case that at least the yeast Mga2 system uses lateral pressure/compressibility as a readout. This finding is of high importance and on its own merits publication.

I'd like to add a few more words: The present manuscript is of remarkable experimental quality and extremely well written. All main conclusions are supported by at least two independent methodological approaches. FRET measurements are backed up by EPR experiments etc. I find the combination of functional readout (Mga2 ubiquitylation) with high-level spectroscopic data extremely appealing. It was a pleasure to carefully read this story and I have to say that it was remarkably difficult to find things that can be improved. I did manage to find some (very minor) points, which are summarized below:

Minor points:

Figures

- Figure 1A: The authors do a very good job of hiding the mechanism they investigate. The membrane lipids (saturated vs. unsaturated), the tryptophan residues, the arrows indicating the rotatory movement of the transmembrane helices are tiny. The transcription factor DNA-interaction instead, which is only context for this manuscript, takes up a lot of space. I'd suggest to rework this artwork in a way that highlights the message of the paper – maybe a generalized scheme of the Ole1 activation with a zoom-in highlighting the membrane in two different states, so that the actual investigated mechanism is visible at first glance.
- Figure 2E,F: Please include n-numbers in the figure legend. I'd also advocate to include a quantification of ubiquitylated product fractions per time point (integrated intensity of product bands divided by remaining intensity of unmodified Mga2 construct, or via densitometric quantification as in Figure 4F). This would give the reader a sense of the variability in the system early on in the manuscript.
- Figure 3: Empty unit brackets should be removed, I'd also recommend to change the headline in figure 4E – acceptor excitation is not FRET, which then results in a comparison of FRET and non-FRET datasets in "FRET in proteoliposomes"
- Figure 4: Empty unit brackets should be removed. Why were different statistical tests used for the FCS data and the other experiments?
- Figure 5: Empty unit brackets should be removed.
- Figure S2: Empty unit brackets should be removed.
- Figure S4: When highlighting double bonds, mark four atoms to highlight stereochemistry - three are actually inconclusive

Experimental suggestions

- I recommend that the authors check the size distributions of the different liposome populations generated with varying lipid compositions, e.g. by DLS, to obtain an idea how much the morphology / number of liposomes is influenced by the lipid composition. I would also check how much the actual lipid content reflects the input, e.g. by re-extraction of the lipids and TLC or MS analysis. This would not necessarily require proteoliposomes (I do not think that the sensor construct influences these parameters given its low abundance). These experiments can be carried out with pure lipid liposomes which should be feasible in a reasonable timeframe.

André Nadler

Reviewer #3:

Remarks to the Author:

The manuscript "Regulation of lipid saturation without sensing membrane fluidity" by Stephanie Ballweg et al discusses the molecular mechanism of regulation of phospholipid saturation in membranes by transmembrane proteins which do not depend on the overall membrane fluidity. The construct used to sense the lipid saturation is quite novel and

has been established by the authors in their earlier studies. However, there are few questions regarding the authenticity of their claims which needs to be clarified before their claims can be accepted. I list some of these points below for careful consideration by the authors:

1. The main idea of this work is based on the premise that the sense-response construct that they have created can detect the abundance, number and location of double bonds in lipid acyl chains in membranes which acts as trigger for production of unsaturated lipids in cells. Further, they hope to distinguish whether Mga2 senses viscosity/fluidity of membranes or specificity of double bonds in acyl chains. Figure 4 is a key data in this regard. First of all the FCS data which the authors claim to distinguish between viscosity of POPC and mixed POPE/POPC based membranes is very ambiguous. In fact the distributions of diffusivity values are so broad (especially for PE based membranes) that it is impossible to claim that those have higher viscosity than POPC membranes. This breadth could be due to the multicomponent and heterogeneous nature of the lipid membrane based on PE.

2. Lot of earlier studies have used DPH dye polarisation as indication of membrane fluidity. This technique could alternatively be used to establish the actual membrane viscosity/fluidity. The FRET studies provide information from deep inside the hydrophobic core while it is possible that the FCS reporter is doing so from, predominantly, the hydrophilic head group region.

3. It is not clear from which region of the membrane the dye is reporting the fluidity or viscosity, as mentioned above, but there are reports which suggests fluidity gradients in eukaryotic membranes
eg. Hirak Chakraborty et al Langmuir 2015 (Depth-Dependent Organization and Dynamics of Archaeal and Eukaryotic Membranes: Development of Membrane Anisotropy Gradient with Natural Evolution). It is possible that the FRET and FCS data could come from different regions of the membrane and might be responsible for viscosity-ubiquitylation anti correlation. Hence this needs to be cross-checked with a dye reporting mobility near the hydrophobic core where FRET reporters are located.

4. Similarly densitometric data extracted from the the SDS-Page and immunoblotting analysis (4E) is very ambiguous and it is hard to tell the difference between PC and PE membranes as far as ubiquitylation is concerned. In Fig 4F the difference is also largely negligible except for 2-3 points in PC based system. The authors use this data to conclude that Mga2 does not report membrane viscosity. However, based on the data presented in Fig 4 this cannot be unambiguously established.

5. The data in Fig 5 is more unambiguous as far as the correlation between viscosity (diffusivity) and FRET efficiency or structure of the sense-response construct is concerned. In Fig 3 also the authors established a clear correlation between degree of saturation as well as viscosity and the FRET efficiency. On the contrary, the ubiquitylation data seems to anti-correlate with FRET efficiency and hence to the structure of the sensor which is in some sense anti-thesis to their claim that the sensor senses local environment or structure of double bonds in lipid acyl chains and not the global membrane mobility. There is of course clear anti-correlation between ubiquitylation and viscosity but then the premise of structure-function correlation is clearly missing.

6. The authors also conclude that their sensor construct senses lateral pressure or compressibility changes in the membrane hydrophobic core. They do not provide any

data in this regard but refer to some possible correlations alluded to by others.
This cannot therefore be a major conclusion of their own work as claimed in the abstract.

In conclusion, the authors have come up with a clever scheme to correlate membrane homeoviscous adaption of eukaryotic cell membranes with the local conformation of the lipid acyl chains. The results if established firmly could enhance our understanding of an important functional pathway of cells. However, given these questions which arise about the nature and interpretation of the data I would be reluctant to accept this manuscript in its current form in Nature Communications

We would like to thank all three reviewers for their constructive and very helpful comments. We feel that the quality of the manuscript has been tremendously improved by this review process.

Reviewers' comments:

Reviewer #1 (Remarks to the Author):

The manuscript by Ballweg and colleagues addresses a very timely and important question in membrane biology, i.e. how is the membrane saturation state sensed. In a previous publication the authors have shown that in yeast the ER-resident membrane protein Mga2 protein acts as a sensor for membrane lipid saturation to maintain a constant degree of membrane fluidity. Based on complementary cellular, in vitro and in silico (MD simulations) studies the authors suggested a transmembrane helix rotation-based mechanism of sensing lipid saturation, with a key role of a tryptophan residue that adopts different orientations depending on the degree of lipid saturation. In essence, a saturated lipid environment stabilizes a dimer conformation with the two tryptophans of each protomer facing towards the dimerization interface, leading to efficient ubiquitylation of the cytosolic protein domain, subsequent proteolytical digest and release of a fragment that in the nucleus activates transcription of the desaturase Ole1. In the current manuscript the authors now advance these studies to elucidate the molecular mechanism underlying Mga2's sensor activity. To this end the authors have established an elegant in vitro assay to measure ubiquitylation of a so-called sense-and-response reporter containing a truncated version of Mga2. Using this assay the authors investigated the effect of different liposomal membrane properties to decipher biophysical parameter, such as viscosity and lateral pressure profiles, for their ability to modulate ubiquitylation efficiency. In addition, a FRET approach was used to correlate ubiquitylation efficiency with proximity of E3 ubiquitin ligase binding site and ubiquitin acceptor site within the cytosolic domain of Mga2. From their data the authors suggest that Mga2 senses the lateral pressure profile at a specific site within the bilayer but not the overall state of membrane viscosity/fluidity. This work significantly adds to our understanding of molecular mechanisms contributing to membrane homeostasis.

We thank reviewer 1 for the positive assessment of our manuscript. We believe that the reviewer's constructive comments lead to a substantial improvement of the manuscript.

Comments

1. As acknowledged by the authors it is a major challenge to study different membrane properties separately due to a high degree of interdependency, which raises some questions concerning the interpretation of the results. To support the model proposed by the authors and to get further insight at a molecular level, the work would benefit from a complementary method such as MD simulations.

Following the reviewer's advice, we have performed extensive MD simulations (**new Figures S6E-H, new Figure S3, Figure 7, and Figure S7**).

Firstly, we have performed all-atom MD simulations of five bilayer systems and employed a particularly powerful way to highlight subtle differences in the packing density of lipids within the bilayer (**new Figure S3, Figure 7 and Figure S7**). Using this approach analyzing the local number density of lipid atoms weighted by the atom size, we provide a new framework to visualize the qualitatively distinct impact of saturated lipid acyl chains and small lipid headgroups (i.e. the headgroup of PE) on the local packing density in the bilayer. These findings explain better how Mga2 can be sensitive to changes in lipid saturation, while being less responsive to changes in the lipid headgroups (i.e. the PC-to-PE ratio). This differential sensitivity is due to highly localized functional sensitivity of the TMH of Mga2 senses in a specific region of the bilayer marked by the position of the sensory tryptophan W1042. In the revised manuscript, we dedicate an entire, new paragraph to these – in our view- important findings.

Secondly, we now compare the structural dynamics of the WT TMH from Mga2 with in total four mutant variants (data for the WT TMH and the W1042A are re-plotted from a previous publication with permission) using extensive coarse-grained MD simulations. Consistent with our model, we find that those mutant variants that impair the activity of Mga2 *in vivo*, have also a strong impact on the structural dynamics of the dimeric TMHs. We now state: “Notably, those mutants with a particularly strong impact on the structural dynamics in the TMH region (W1042F and W1042A) also exhibit a strong functional defect in vivo (Figure 6A-D, **new Figure S6E-H**).”

2. Could the author comment on potential structural requirements to the TM domains to sense changes in lateral pressure profiles – such as hourglass shaped TM structures.

We are grateful to the reviewer for highlighting this important point, which encouraged an intense discussion with various experts in membrane biophysics.

Our revised manuscript is based on an intense exchange with Gerhard Hummer (MPI Biophysics), Roberto Covino (MPI Biophysics), Alex Sodt (NIH), Edward Lyman (University of Delaware), Ilya Levental (UTHealth Houston), and Milka Doktorova (UTHealth, Houston). As a consequence, we do not favor a central role of the lateral pressure profile for controlling the activity of the lipid saturation sensor Mga2 any longer. We now dedicate an entire paragraph and the **new Figure S3, Figure 7, and Figure S7** to this important point. We have rewritten substantial parts in the introduction and the discussion accordingly.

In the revised manuscript, we now determine the lateral pressure profile for five different bilayer systems (**Figure S3C**) thereby allowing for a cross-validation of functional data from the *in vitro* experiments with the results from MD simulation.

We would like to stress that the lateral pressure profile, the lateral compressibility profile, and the local number density of lipid atoms are to some degree related, but clearly distinct. Possibly surprising for the non-expert readers, we find that there is no correlation between the local packing density (as determined by the local number density of lipid atoms) and the lateral pressure in some regions of the membrane

including the region between 3 and 10 Å from the bilayer center where the sensory W1042 residue is located (compare trends in **Figure S3C** and **Figure 7**). .

As suggested by the reviewer, we now comment on the role of hourglass-shaped proteins as potential sensors of curvature stress and/or the lateral pressure profile and compare those to rather cylindrical, dimeric TMHs of Mga2.

3. Due to the strong interdependency of biophysical parameters dictating membrane properties, I think the number and type of different conditions chosen are not yet entirely convincing to support the authors conclusion that only lateral pressure is sensed by Mga2. At this stage I would rather conclude that both parameters are sensed.

The reviewer raises an important issue and we could not agree more. Identifying a SINGLE membrane property as the central measured variable of Mga2 is not feasible at the moment and quite possibly will never be.

Nevertheless, our work rules out a critical role of membrane fluidity/viscosity in regulating lipid saturation in baker's yeast. For the revised manuscript, we refined our discussion of the possible underlying mechanism(s).

We have added a substantial amount of data and employed a powerful, analytic tool to highlight differences in the local packing densities within a bilayer (**new Figure 7, S7**). Based on new findings, we favor an important role of the local packing density and/or the lateral compressibility profile, which are sensed by the bulky TMH residue W1042 in Mga2.

We are convinced that our study provides an important step to better understand the impact of collective bilayer properties on the structure, localization, and function of membrane proteins, by providing a blueprint of how to approach such challenging questions.

As an example, the authors argue that PE is a factor perturbing membrane viscosity. Increasing amounts of PE were shown not to effect cwERP values, taken as one indicator that viscosity is not parameter sensed by Mga2. However, PE was reported to not only affect membrane viscosity but also the lateral pressure profile. Can the authors rule out a contribution to lateral pressure profiles?

The reviewer is right. We cannot exclude a contribution of the lateral pressure profile. Based on all-atom MD simulations, we have determined the lateral pressure profile of five bilayer systems including a PE-containing bilayer (**new Figure S3**). Expectedly, we find characteristic features in the lateral pressure distribution inside the bilayer, caused by the presence of PE lipids. While we cannot entirely rule out an impact of the lateral pressure profile on the activity of Mga2, we favor a different model, which is -in our view- strongly supported by new data.

Somewhat in line with the reviewer's concern, we find that the inclusion of PE lipids in a PC-based bilayer system has an impact not only on the lateral pressure profile, but also on a number of other bilayer properties (**new Figure S3**) including the lipid packing inside the membrane (**new Figure 7, S7**). However, we find that changes in lipid

saturation have a stronger and more localized effect on the packing density in the region of the sensory tryptophan W1042 of Mga2 (**new Figure 7, S7**), while the inclusion of PE lipids causes less pronounced and more delocalized effects (**new Figures 7, S7**). We dedicate an entire paragraph in results section of the revised manuscript to this important point and have revised both the introduction and discussion accordingly.

In addition to the cwERP data, proximity by FRET, C-Laurdan, lipid diffusion and ubiquitylation data should be shown for the PE titration experiments to support the authors conclusion.

According to the reviewer's suggestion, we have performed additional experiments with liposomes containing different concentrations of PE (at 0 mol%, 20 mol% and 40 mol% whilst maintaining the overall lipid acyl chain composition constant) (**new Figures S4B, S4C, S4E, S4G, S4I**). For these analyses, we have decided against a bilayer containing 10 mol% of PE, because the impact of the 20 mol% of PE was already very low. We also performed additional MD simulations with a PE-containing bilayer (**new Figures S3, 7, S7**).

Shortly, our C-Laurdan data show that PE lipids increase the lipid packing density of a PC-based bilayer (**new Figure S4C**), which is further supported by MD simulations (**new Figure S3**). FCS data suggest that diffusion of a fluorescent lipid analogue is barely affected by 20 mol% of PE, but significantly affected by 40 mol% of PE in a PC-based lipid matrix (**new Figure S4E**). DPH anisotropy data show that the rotational mobility of this probe is barely affected even at 40 mol% PE (**new Figure S4B**). This is in line with previous studies and suggests that C-Laurdan spectroscopy and lipid diffusion measurements via FCS are more sensitive to changes of the membrane composition than DPH anisotropy (as discussed in the revised manuscript). These findings highlight the explicit sensitivity of the Mga2 system. Importantly, we also performed additional FRET experiments (**new Figure S4G**) and *in vitro* ubiquitylation experiments (**new Figure S4I**), which are fully consistent with our previous findings and statements that Mga2 is rather selectively affected by changes of lipid saturation and much less so by changes in the lipid headgroup composition. We discuss these findings in light of the insights from MD simulations (**new Figure S3, Figure 7, Figure S7**) in the revised manuscript. Furthermore, we have substantially increased the number of individual experiments for several panels in Figure 4 and 5 to further increase the statistical power of our analyses.

Temperature-dependent changes in lipid profiles in *S. cerevisiae* were reported to significantly alter the PE:PC ratio. Can the authors comment on possible roles of lipid headgroups in Mga2's sensor function?

In the revised manuscript, we dedicate an entire new paragraph to discuss the possible role of lipid headgroups on Mga2's sensor function. While our data are consistent with a modulatory function of the lipid headgroup composition, we find an explicit sensitivity of the Mga2 sensor to the lipid acyl chain composition and discuss our findings in the revised manuscript more extensively.

4. The data on amino acid requirements at the sensor site is intriguing. How do these variants behave in the FRET proximity and ubiquitylation assay, especially concerning

the F,Y and Q variants? Do these variants show the same dependency on membrane properties as the wildtype (e.g. comparison of DOPC vs POPC).

We are happy that the reviewer shares our fascination and enthusiasm regarding the mutagenesis of the sensory tryptophan residue. As stated above, we have performed extensive molecular dynamics simulations regarding the mutant variants mentioned by the reviewer (**new Figure S6-E-H**). Consistent with a central role of the sensory tryptophan W1042, we find a major impact of the W1042F, W1042Y, and W1042Q mutations on the structural dynamics of the TMH.

However, considering the relatively short time for the revision, the number of additional experiments performed, and in light of the challenging quantification of high-molecular poly-ubiquitylation smear from immunoblots that gives rise to a relatively poor signal-to-noise ratio, we have not performed additional ubiquitylation experiments for other mutant variants, which should fall between WT and W1042A (**see new Figure S6C, S6D**). We have also decided against additional FRET experiments, because the FRET efficiency cannot predict the ubiquitylation efficiency (see our answer to point 5 of reviewer 1).

5. Lines 333ff, comparing Figure 4 and 5: Can the authors comment on the correlation of proximity by FRET versus ubiquitylation efficiency.

The reviewer highlights an important point (see also our answer to reviewer 3 point 5). We have carefully rephrased the respective section in the manuscript. In fact, a single FRET pair (providing a single, average FRET efficiency) cannot unambiguously describe the entire structural dynamics of the juxta-membrane domain moving in time and space. While an increased FRET efficiency implies an increased average proximity of the two fluorophores, it does not predict a perfect distance and orientation of the lysine residues targeted by the E3-ubiquitin ligase for ubiquitylation. We use the FRET assays to establish a structural coupling between the TMH and juxta-membrane residues, but do not use the FRET assay for predicting ubiquitylation. In the revised manuscript, we now make this point clearer.

We now comment on the (partial lack-of-correlation) between FRET proximity versus ubiquitylation efficiency by stating: “Furthermore, we find that the FRET efficiencies of the K969A and K983D pair in different bilayers (Figure 5E, 5F, S5G, S5H) do not correlate with the respective ubiquitylation efficiencies (Figure 5G, 5H, S5I). This is not entirely surprising, because a single FRET pair cannot describe the entire structural dynamics in the region of ubiquitylation. While increased FRET efficiencies imply an increased average proximity of the two fluorophores, it does not imply a perfect distance and relative orientation of the target lysine residues K980, K983 and K985 and the E3-ubiquitin ligase Rsp5.”

Regardless of these arguments, we consider it extremely important to show that there is structural coupling between the TMH and the juxta-membrane region as we have done.

6. Lines 343ff: Can the authors comment on the biological implication, i.e. range of db

positions in S.c. membranes? How does delta9-C16:1 cis/trans behave?

Following the reviewer's suggestion, we have commented on the range of double bond positions and the abundance of unsaturated lipid acyl chains in *Saccharomyces cerevisiae*. We now state: "Baker's yeast can synthesize only $\Delta 9$ -cis mono-unsaturated fatty acids, and the average proportion of these unsaturated acyl chains lies between 65% and 75% in glycerophospholipids depending on the temperature of cultivation⁸. The sense-and-response construct cannot be ubiquitylated in a relatively unsaturated membrane (75% $\Delta 9$ -cis 18:1 acyl chains), but it is robustly ubiquitylated in POPC, which is slightly more saturated (50% $\Delta 9$ -cis 18:1 acyl chains) (Figure 2F)."

As suggested by the reviewer, we have performed additional experiments using PC C16:1 $\Delta 9$ -*trans* PC and C16:1 $\Delta 9$ -*cis* PC, even though PC C16:1 $\Delta 9$ -*trans* PC is not even found in *Saccharomyces cerevisiae*. Unfortunately, the Mga2 sensor could not be reconstituted efficiently in a C16:1 $\Delta 9$ -*cis* PC bilayer. Nevertheless, we have characterized in depth the C16:1 $\Delta 9$ -*trans* PC bilayer with various spectroscopic techniques and studied its impact on the Mga2 sensor (**new Figures S5C-I**). Most importantly, the C16:1 *trans* PC bilayer mimics a rather saturated membrane and supports the ubiquitylation of the sense-and response construct by Rsp5 (**new Figure S5I**). This finding further underscores a central role of the acyl chain configuration on Mga2 activation.

7. Are the Mga2 reconstitution efficiencies comparable for the various conditions used to generate proteoliposomes?

We have tested the reconstitution efficiencies and their yields are comparable for all tested lipid compositions (with C16:1 $\Delta 9$ -*cis* PC bilayer being an exception as stated above). We have rephrased the relevant section in the manuscript. We now state: "The yield of the reconstitution was tested after harvesting the proteoliposomes and it was comparable for all the desired protein and lipid compositions used in this study."

8. Line 163: Only a protein:lipid ratio of 1:8,000 is shown in that Figure.

We thank the reviewer for spotting this inaccuracy. We have corrected the relevant section. We have now correctly state the protein-to-lipid ratio in the legends for Figure S1D, S2H, and S5H (including in part newly performed sucrose gradients).

9. Figure 2E: A protein-to-lipid ratio of 1:5,000 is used here but in most other experiments it is 1:8,000. I would suggest to use the same protein-to-lipid ratio in same types of experiments and move comparisons of different ratios into the Supplement.

We have used a higher protein-to-lipid ratio of 1:5,000 for these important control experiments to increase the sensitivity to even most modest levels of background ubiquitylation of the control constructs (deltaLPKY and 3KR). We do not see substantial ubiquitylation of these control constructs even under these conditions.

In the revised manuscript we now explain our rationale for this decision more clearly. We now state: "Notably, these experiments were performed at a relatively high protein-

to-lipid ratio of 1:5,000 to increase the sensitivity for a potential background ubiquitylation of the control constructs.”

In FRET experiments, we consistently used a lower protein-to-lipid ratio of 1:8,000 in order to minimize the signal from proximity FRET. This is FRET signal from labeled molecules, that exchange energy from the donor to the acceptor only due to an occasional proximity of the two labeled proteins ‘bumping’ into each other while freely diffusing in the proteoliposomes. We now state: “We used a low protein-to-lipid ratio of 1:8,000 in these experiments to minimize the contribution of unspecific proximity FRET to the overall signal⁴⁵”.

10. Figure 3G: filling of bar graph could be changed to white to fit with Figure 4 data.

We have followed the reviewer’s advice.

11. Figure 4A: I would suggest to replace the arrow indicating increasing viscosity by a triangle as used in B for describing the extent of packing density.

We have followed the reviewer’s advice for Figure 4A and Figure 5A.

12. Figure 4B: Check x-axis labeling

Following the advice, we have moved the labels in the x-axis in the proper position and added additional x-axis labeling.

13. Figure S4C: The author should comment on the strikingly different properties of liposomes (used for the experiments described in Figure 5) in sucrose gradient centrifugations.

See our **new Figure S5F**. We perform the sucrose gradient primarily to exclude the massive aggregation of the sensor protein in the course or reconstitution. If this would happen, we would expect a substantial signal of the sensor protein in fraction 13 of the gradient.

We have added additional experiments and have repeated several sucrose gradient centrifugations (**new Figure S5F**). We do not fully understand the behavior of our proteoliposomes in the sucrose gradient, but we are currently performing systematic experiments with protein-free liposomes and with different proteoliposomes to reach a better understanding.

In the revised manuscript we now state: “To this end, we used our newly established FRET reporter (K983D+K969A) and reconstituted it successfully in different membrane environments as judged by sucrose density gradients (Figure S5F). The reason for the different buoyant densities of the proteoliposomes remains to be fully explored.” We suspect that in some lipid compositions there may be a larger heterogeneity of the protein:lipid ratio from proteoliposome to proteoliposome. We also speculate that the lipid packing density (area per lipid; see **new Figure S3A**) might directly affect the

effective lipid volume and thus, the buoyant density of a liposome. We are currently in the process of exploring this possibility more systematically.

14. Check legends to Figure 4 and 5: data refer to Figure 3F-G not 2F-G.

We have corrected this error.

15. Figure 4-5: The authors describe diffusion coefficients of lipids in the in vitro setting. These data should be discussed in comparison to values obtained in cellular membranes/in presence of transmembrane domain, i.e. how does the range of viscosity tested relate to the in vivo situation.

In response to the reviewer's question, we have extended the discussion of this point. *In vivo*, the viscosity is influenced by several parameters such as lipid composition, protein composition, cell wall, cytoskeleton etc. It is not clear how each component affects the viscosity. However, the diffusivity is known to be an order of magnitude faster in in vitro GUV systems than in cellular systems. In the intermediate cell models such as giant plasma membrane vesicles, it is also 3-5 times faster (see for example Pinkwart et al, JBC, 2019; Schneider et al, MBoC, 2017).

We now state: "Notably, this lipid diffusion coefficient is roughly an order of magnitude lower than in cellular membranes⁴⁶ and also lower ~3-5-fold than in isolated giant plasma membrane vesicles⁴⁷."

16. Figure 6E: Why was the protein:lipid ratio changed to 1:15,000 as compared to 1:8,000 as used in the experiments shown in the main text figures?

We have performed these experiments at a protein-to-lipid ratio of 1:15,000 in an attempt to be less wasteful. These experiments were performed before the other experiments at the different ratio of 1:8,000 had been performed. For the revised manuscript, we have moved these data in the supplementary materials to avoid irritations due to an inconsistency of the protein-to-lipid ratio in these experiments (**now Figure S6C,D**).

17. In order to perform the cwERP measurements the authors had to use a protein:lipid ratio of 1:500. How does this affect the protein/read out parameters?

In the revised manuscript, we now mention the relatively high protein-to-lipid ratio more explicitly by stating: "The relatively high protein-to-lipid ratio of 1:500 required for these EPR experiments render the resulting proteoliposomes a bit more similar to cellular membranes, which exhibit protein-to-lipid ratios of ~1:80¹⁸."

The precise impact of membrane proteins on the lipid bilayer properties is a matter of active research. It is expected that membrane proteins affect most, if not all, membrane properties. The density of transmembrane segments in these cwEPR experiment is still ~5-10 fold lower than in native membrane. Future experiments in our lab shall address the role of membrane proteins, their crowding in cellular membranes, and

more complex lipid compositions on the Mga2 system and other membrane property sensors.

18. Can the authors exclude that the presence or absence of the Leucine zipper does not influence dynamics of the TM helices?

We do not think that the zipper has a substantial impact on the freedom of the TMHs in the dimer to rotate against each other. The zipper is separated from the TMH by a flexible linker, the globular Maltose binding protein of ~50 kDa, another flexible linker including a protease recognition site, and the juxtamembrane domain of Mga2. We thus think that the TMHs can rotate freely in the membrane. Furthermore, the zipper functionally resembles the IPT domain, which is found in the native, full-length Mga2: The IPT domain keeps Mga2 in a dimeric form (Piwko and Jentsch, PMID: 16845392). The leucine zipper in our bottom-up construct therefore resembles a structural feature of functional significance, which is also found in the original, full-length Mga2.

19. Line 396: unique sensitivity...of...

We added the missing word.

20. Line 1043: check sentence

We corrected this part of the manuscript.

Reviewer #2 (Remarks to the Author):

Stephanie Ballweg et al. address one of the key questions in membrane biology in their manuscript: How do cells measure the unsaturation degree in their membranes? It has often been assumed that cellular sensors react to changes in membrane viscosity, which is correlated with unsaturation degree, but the actual data basis for this assumption has been remarkably weak. The authors now make a convincing case that at least the yeast Mga2 system uses lateral pressure/compressibility as a readout. This finding is of high importance and on its own merits publication. I'd like to add a few more words: The present manuscript is of remarkable experimental quality and extremely well written. All main conclusions are supported by at least two independent methodological approaches. FRET measurements are backed up by EPR experiments etc. I find the combination of functional readout (Mga2 ubiquitylation) with high-level spectroscopic data extremely appealing. It was a pleasure to carefully read this story and I have to say that it was remarkably difficult to find things that can be improved. I did manage to find some (very minor) points, which are summarized below:

We would like to thank the reviewer for his very positive assessment of our manuscript.

Minor points:

Figures

- Figure 1A: The authors do a very good job of hiding the mechanism they investigate. The membrane lipids (saturated vs. unsaturated), the tryptophan residues, the arrows indicating the rotatory movement of the transmembrane helices are tiny. The transcription factor DNA-interaction instead, which is only context for this manuscript, takes up a lot of space. I'd suggest to rework this artwork in a way that highlights the message of the paper – maybe a generalized scheme of the Ole1 activation with a zoom-in highlighting the membrane in two different states, so that the actual investigated mechanism is visible at first glance.

We have optimized the schematic representation in Figure 1A to better highlight the central topic(s) of this manuscript. Furthermore, we added a **new Figure 7**, which might further help the reader to grasp the essence of the mechanism employed by Mga2.

- Figure 2E,F: Please include n-numbers in the figure legend.

In the revised manuscript, we now provide the n-numbers in the figure legends.

I'd also advocate to include a quantification of ubiquitylated product fractions per time point (integrated intensity of product bands divided by remaining intensity of unmodified Mga2 construct, or via densitometric quantification as in Figure 4F). This would give the reader a sense of the variability in the system early on in the manuscript.

We now provide information the requested information in Figure 2E, F.

- Figure 3: Empty unit brackets should be removed, I'd also recommend to change the headline in figure 4E – acceptor excitation is not FRET, which then results in a comparison of FRET and non-FRET datasets in “FRET in proteoliposomes”

According to the reviewer's suggestion, we removed the empty brackets.

We decided to keep the title in Figure 3E to guide the reader's eye in that the data in Figure 3E are not derived from samples in detergent solution as the previous data in Figure 3B-D, but from proteoliposomes.

- Figure 4: Empty unit brackets should be removed. Why were different statistical tests used for the FCS data and the other experiments?

According to the reviewer's suggestion, we removed the empty brackets. The number of experimental values for FCS measurements is usually higher than the other type of experiments (>50 for FCS vs 1-10 for others). Kolmogorov-Smirnov test is a type of unpaired nonparametric t-test. Normal parametric t-test would overestimate the differences in larger samples (due to the denominator s/\sqrt{n}). Kolmogorov-Smirnov test is useful and a general nonparametric method for comparing two samples with large data sets, as it is sensitive to differences in both location and shape of the empirical cumulative distribution functions of the two samples.

- Figure 5: Empty unit brackets should be removed.

According to the reviewer's suggestion, we removed the empty brackets.

- Figure S2: Empty unit brackets should be removed.

According to the reviewer's suggestion, we removed the empty brackets.

- Figure S4: When highlighting double bonds, mark four atoms to highlight stereochemistry - three are actually inconclusive

This is correct. We have followed the reviewer's suggestion.

Experimental suggestions

- I recommend that the authors check the size distributions of the different liposome populations generated with varying lipid compositions, e.g. by DLS, to obtain an idea how much the morphology / number of liposomes is influenced by the lipid composition.

As recommended by the reviewer, we have performed DLS experiments and find that all liposomes have diameters between 61 nm and 143 nm. PC-based liposomes were in the range of 61 nm to 72 nm, while the size distribution of PE based liposomes was broader and the liposomes on average larger (143 nm). We mention this in the revised manuscript. Notably, our optimized, current protocol for the reconstitution of the sense-and-response construct starts from fully detergent-solubilized mixtures of proteins and lipids. We are currently in the process of developing alternative reconstitution protocols for a more rigid control of the resulting size of proteoliposomes.

The different sizes of the proteoliposomes raise the question, if membrane curvature might impact on the ubiquitylation efficiency. Especially, it raises the question if the lack of sensor ubiquitylation in PE-containing membranes (**Figure 4E, 4F, new Figure S4I**) is due to membrane curvature. We doubt this is the case.

We doubt that the observed membrane curvatures manifest substantially at the molecular scale. However, we are aiming to address the role of more extreme membrane curvatures (<50 nm) on the activity of Mga2 in the near future in greater detail using a combination of experiments and MD simulations. Addressing the role of membrane curvature on all aspects of membrane biophysics and the structural dynamics of Mga2 is beyond the scope of this study.

In line with our answer to reviewer 1, we cannot and do not exclude an impact of PE on membrane curvature stress and/or the local lipid packing. In fact, we show that PE-containing bilayers have different lateral pressure profiles than PE-free ones (**new Figure S3C**). Our MD simulations suggest that PE also has an impact on the

distribution of the number density of lipid atoms (**new Figure 7 and Figure S7**) as well as other bilayer properties (**Figure S3A,B and D**).

I would also check how much the actual lipid content reflects the input, e.g. by re-extraction of the lipids and TLC or MS analysis. This would not necessarily require proteoliposomes (I do not think that the sensor construct influences these parameters given its low abundance). These experiments can be carried out with pure lipid liposomes which should be feasible in a reasonable timeframe.

We have used TLC analysis to study the lipid composition of proteoliposomes after reconstitution (**new Figure S4F**). Expectedly, we find PE in proteoliposomes generated from a PE-containing lipid mixture.

Reviewer #3 (Remarks to the Author):

The manuscript "Regulation of lipid saturation without sensing membrane fluidity" by Stephanie Ballweg et al discusses the molecular mechanism of regulation of phospholipid saturation in membranes by transmembrane proteins which do not depend on the overall membrane fluidity. The construct used to sense the lipid saturation is quite novel and has been established by the authors in their earlier studies. However, there are few questions regarding the authenticity of their claims which needs to be clarified before their claims can be accepted. I list some of these points below for careful consideration by the authors:

1. The main idea of this work is based on the premise that the sense-response construct that they have created can detect the abundance, number and location of double bonds in lipid acyl chains in membranes which acts as trigger for production of unsaturated lipids in cells. Further, they hope to distinguish whether Mga2 senses viscosity/fluidity of membranes or specificity of double bonds in acyl chains. Figure 4 is a key data in this regard. First of all the FCS data which the authors claim to distinguish between viscosity of POPC and mixed POPE/POPC based membranes is very ambiguous. In fact the distributions of diffusivity values are so broad (especially for PE based membranes) that it is impossible to claim that those have higher viscosity than POPC membranes. This breadth could be due to the multicomponent and heterogeneous nature of the lipid membrane based on PE.

We appreciate the reviewer's concern on this delicate point and have added more FCS data to **Figure 4A, 4D, 4F, 5A, 5F and 5H** for better statistics. We also performed additional DPH anisotropy measurements (**new Figure S4B, and S4E**). Furthermore, we extended our discussion and highlight the fact that the differences in membrane fluidity for some of the lipid compositions in this study are rather low.

We fully agree with the reviewer that the differences of membrane fluidity imposed by the (physiologically relevant) changes in the lipid acyl chain composition are very low. In fact, we now show based on newly added data that these changes of fluidity are not even detected by DPH anisotropy (**new Figure S4B, and S5E**). Most strikingly, in our view, is the fact that these changes of the lipid acyl chain composition are still sufficient to switch the activity of the Mga2 sensor.

In order to address maybe some of the concerns raised by the reviewer, we have added many more FCS data for better statistics. Obviously, FCS relies on the diffusion time of fluorescent molecules in a small confocal volume. It is a method with single-molecule sensitivity/resolution, which has an inherently high error (looking at individual fluorescent events). Moreover, for multi-component mixtures, the breadth of the data is larger due to the inherent heterogeneities of the sample. For these reasons, for this type of experiments, large amount of data should be collected for proper statistics. Consistently, the errors in our measurements are within the usual range for this methodology and the statistical tests suggest that the diffusivity between the chosen lipid compositions is indeed distinct. This difference cannot be caused by multi-component diffusion because that scenario would yield different type of curves where two components would be evident.

We would like to stress that the lipid compositions have not been chosen to yield very high difference in membrane fluidity. Instead, we have chosen them to reflect physiologically relevant differences in the lipid acyl chain composition. We find it quite exciting that Mga2 can sense these (indeed rather subtle) variations.

Overall, we appreciate the reviewer's concern that the differences are small and that the breadth of the data is wide. For the revised manuscript, we have performed a large number of additional FCS experiments for even better statistics. The number of experimental data and statistical tests suggest that PE causes slight but significant viscosity change (in line with our original statements and with the published literature). Despite the differences being small, Mga2 responds. This highlights the remarkable sensitivity of Mga2.

2. Lot of earlier studies have used DPH dye polarisation as indication of membrane fluidity. This technique could alternatively be used to establish the actual membrane viscosity/fluidity. The FRET studies provide information from deep inside the hydrophobic core while it is possible that the FCS reporter is doing so from, predominantly, the hydrophilic head group region.

Following the reviewer's advice, we have performed additional experiments using DPH anisotropy (**new Figure S4B and S5E**). These newly added DPH anisotropy data have helped us a lot to highlight the explicit sensitivity of the Mga2 system.

Regarding the data in Figure S4B we now state: "Intriguingly, the anisotropy of the membrane-probe DPH is barely affected by such changes of the bilayer composition over a broad range of temperatures (Figure S4B). This indicates that subtle changes of membrane fluidity are detected more sensitively by analyzing lipid diffusion rather than by the anisotropy of the smaller DPH probe."

Beyond that, we fear that our FRET and FCS experiments were misunderstood. We have clarified the relevant sections in the revised manuscript.

We use FRET to study the structural dynamics of Mga2 OUTSIDE the membrane in the region where the ubiquitylation of Mga2 occurs. The fluorescent labels are thus placed in polar regions of the protein far from the membrane.

The FCS reporter is used to determine the diffusion coefficient of the labeled lipids (the lateral, transversal motion of the entire lipid). It does not report on a certain transverse region of the membrane (e.g. the hydrophilic headgroup region).

However, fully in line with the reviewer's important point that the local viscosity could be distinct in different regions of the membrane, we have performed additional MD simulations (**new Figure 7 and S7**). In fact, we find that the number density of lipid atoms differs along the normal of the bilayer. We dedicate an entire paragraph to this important point in the revised manuscript and have clarified our statements in the introduction and the discussion accordingly.

3. It is not clear from which region of the membrane the dye is reporting the fluidity or viscosity, as mentioned above, but there are reports which suggests fluidity gradients in eukaryotic membranes eg. Hirak Chakraborty et al Langmuir 2015 (Depth-Dependent Organization and Dynamics of Archaeal and Eukaryotic Membranes: Development of Membrane Anisotropy Gradient with Natural Evolution). It is possible that the FRET and FCS data could come from different regions of the membrane and might be responsible for viscosity-ubiquitylation anti correlation. Hence this needs to be cross-checked with a dye reporting mobility near the hydrophobic core where FRET reporters are located.

We fully agree with the reviewer that certain local properties at different depths in the bilayer exist. In fact, we added new figures that specifically address and highlight this point (**new Figure 7 and Figure S7**). We dedicate an entire paragraph in the results section of the revised manuscript to this centrally important point.

The FRET and FCS data do not come from different regions from within the membrane: FRET data report on the structural dynamics at the site of ubiquitylation (outside the membrane) and FCS reports on the translational movement of lipids molecules.

In response to the reviewer's statement, we also have added DPH anisotropy data (**new Figure S4B, new Figure S5E**). In the revised manuscript, we now discuss the difference of diffusion measurements via FCS and DPH anisotropy for analyzing membrane fluidity, respectively. See also our answer to reviewer 3 point 2.

Consistent with our original statements, our new data highlight the explicit sensitivity of the Mga2 system to increased lipid saturation. We have carefully rephrased all relevant sections in the manuscript.

4. Similarly densitometric data extracted from the the SDS-Page and immunoblotting analysis (4E) is very ambiguous and it is hard to tell the difference between PC and PE membranes as far as ubiquitylation is concerned. In Fig 4F the difference is also largely negligible except for 2-3 points in PC based system. The authors use this data to conclude that Mga2 does not report membrane viscosity. However, based on the data presented in Fig 4 this cannot be unambiguously established.

In response to the reviewer's concerns, we have added substantial new data and performed additional *in vitro* ubiquitylation experiments to improve the statistics (e.g.

in **Figure 4F, 5H, new Figures S4I, S5I**). However, we are aware that it is technically challenging to quantify poly-ubiquitylation ‘smear’ especially based on immunoblotting. This is a well-known fact in the field of protein ubiquitylation.

Being aware of this challenge, we have performed a large number of additional, independent *in vitro* ubiquitylation experiments and performed statistical tests without omitting any data-points. For the revised manuscript, we have performed many more *in vitro* ubiquitylation experiments and improved the statistics (**Figure 4F, 5H, S4I, S5I**). We are convinced that the variability of the *in vitro* ubiquitylation assay comes predominantly from the variability of the detection method (via immunoblotting) and not from a variability of the proteoliposomes or other factors. Despite this, our data show that the robust degree of ubiquitylation observed in POPC membranes (**Figure 4F**) is not observed in membranes containing 40mol% POPE (**Figure 4F**) as supported by statistical tests.

5. The data in Fig 5 is more unambiguous as far as the correlation between viscosity (diffusivity) and FRET efficiency or structure for the senso-response construct is concerned. In Fig 3 also the authors established a clear correlation between degree of saturation as well as viscosity and the FRET efficiency. On the contrary, the ubiquitylation data seems to anti-correlate with FRET efficiency and hence to the structure of the sensor which is in some sense anti-thesis to their claim that the sensor senses local environment or structure of double bonds in lipid acyl chains and not the global membrane mobility. There is of course clear anti-correlation between ubiquitylation and viscosity but then the premise of structure-function correlation is clearly missing.

The reviewer identifies an important point. We agree that there is a lack of correlation between the FRET efficiency and the ubiquitylation efficiency. In the revised manuscript, we address this important point. (See our answer to reviewer 1 point 5).

Shortly, while an increased FRET efficiency does imply an increased average proximity of the two dyes forming the FRET pair, it does not predict a suitable distance and orientation of the lysine residues relative to the E3-ubiquitin ligase for efficient ubiquitylation. In the revised manuscript we now state: “Furthermore, we find that the FRET efficiencies of the K969A and K983D pair in different bilayers (Figure 5E, 5F, S5G, S5H) do not correlate with the respective ubiquitylation efficiencies (Figure 5G, 5H, S5I). This is not entirely surprising, because a single FRET pair cannot describe the entire structural dynamics in the region of ubiquitylation. While increased FRET efficiencies imply an increased average proximity of the two fluorophores, it does not imply a perfect distance and relative orientation of the target lysine residues K980, K983 and K985 and the E3-ubiquitin ligase Rsp5.”

Nevertheless, we consider the FRET assay as very important. It identifies and documents a structural coupling between the TMH and juxta-membrane region. In response to the comments of reviewer 1 and 3, we have thus clarified our statements in the revised manuscript.

6. The authors also conclude that their sensor construct senses lateral pressure or

compressibility changes in the membrane hydrophobic core. They do not provide any data in this regard but refer to some possible correlations alluded to by others. This cannot therefore be a major conclusion of their own work as claimed in the abstract.

We have carefully rephrased our statements in the abstract throughout the manuscript to avoid too strong claims.

For the revised version, we have performed extensive MD simulations and have employed an unconventional approach to represent the molecular packing in lipid bilayers and differences between them (**new Figure 7 and Figure S7**).

In conclusion, the authors have come up with a clever scheme to correlate membrane homeoviscous adaption of eukaryotic cell membranes with the local conformation of the lipid acyl chains. The results if established firmly could enhance our understanding of an important functional pathway of cells. However, given these questions which arise about the nature and interpretation of the data I would be reluctant to accept this manuscript in its current form in Nature Communications

We feel that the additional data (experimental and theoretical) as well as the extended discussion of our findings have substantially improved the quality of the manuscript.

Again, we would like to thank all three reviewers for their time and efforts.

Reviewers' Comments:

Reviewer #1:

Remarks to the Author:

In an impressively short time the authors have managed to add a significant set of new data that further support the proposed model of how in yeast lipid saturation is sensed and regulated by the transcriptional regulator Mga2. The authors now include molecular dynamics simulations as a complementary approach to further investigate how the sensory tryptophan residue within the membrane bilayer can sense and respond to changes in the overall lipid saturation state of the membrane. These data are presented as a new chapter of the results part. In addition, a number of changes were made that altogether significantly improved the manuscript.

I have only a few minor suggestions the authors might want to consider:

- Abstract: The authors should include that MDS studies were performed in addition to the mentioned approaches.
- Data from Figure S3: Results are presented in the legend but not in main text.
- Liposome size, as requested by R2 and measured by the authors, should be included and discussed (as done in the rebuttal letter)
- I assume reviewer #2 meant to quantitatively assess the amounts of the different lipid in liposomes as compared to the starting material
- Typo: Legend to Figure S4F (line 335, concentration)

Reviewer #2:

Remarks to the Author:

The manuscript by Ballweg et al. has been significantly improved. Considering that the original version was already an great paper, this is quite impressive. Frankly, the authors have gone far beyond what was requested and I recommend acceptance of the manuscript in its present form. The concept that cellular membrane property sensors might be measuring local lipid atom packing densities within the bilayer is novel and could serve as a starting point for a new way of analyzing biological membranes.

A few minor recommendations:

- Please check the semantics of the sentence about lipid diffusion coefficients in line 268-270. Currently this seems to state that lipid diffusion in model membranes is an order of magnitude lower than in cellular membranes.
- Remove typos in line 389
- "points" in line 544 as to be "point"
- line 548: should be "has a significant impact on the structural dynamics..."

Reviewer #3:

Remarks to the Author:

I think the authors have provided reasonably satisfactory response to the comments. Also, performed additional experiments/simulations to establish their claims. There are some grey areas but given the challenge at hand and the significance of the results I would be happy to accept this manuscript in Nature Communications. I hope this work generates further interest in this important field and leads to deeper understanding of membrane biophysics of sensing of

membrane fluidity and lipid saturation leading to various signalling pathways for cellular response to external stress.

Reviewer #1 (Remarks to the Author):

In an impressively short time the authors have managed to add a significant set of new data that further support the proposed model of how in yeast lipid saturation is sensed and regulated by the transcriptional regulator Mga2. The authors now include molecular dynamics simulations as a complementary approach to further investigate how the sensory tryptophan residue within the membrane bilayer can sense and respond to changes in the overall lipid saturation state of the membrane. These data are presented as a new chapter of the results part. In addition, a number of changes were made that altogether significantly improved the manuscript.

We thank the reviewer for the positive assessment and for the helpful, additional comments.

I have only a few minor suggestions the authors might want to consider:

- Abstract: The authors should include that MDS studies were performed in addition to the mentioned approaches.

We now mention molecular dynamics simulations in the abstract.

- Data from Figure S3: Results are presented in the legend but not in main text.

The reviewer is correct. We only refer to the data from Figure S3 in the text, but we do not discuss them extensively. We provide the characterization of molecular dynamics simulation data predominantly as a reference and quality control. All data presented in Figure S3 are in line with published literature. We feel that an exhaustive discussion of these data would distract the reader from the main messages of the manuscript.

- Liposome size, as requested by R2 and measured by the authors, should be included and discussed (as done in the rebuttal letter)

We have included the data and a discussion of the liposome size in the manuscript (new Supplementary Figure S4F).

- I assume reviewer #2 meant to quantitatively assess the amounts of the different lipid in liposomes as compared to the starting material

A quantitative assessment using for example lipid mass spectrometry was not possible in the short time of the rebuttal. However, we do not have any indication that certain lipids got selectively depleted during the reconstitution procedure. This is supported by more recent C-Laurdan spectroscopy data performed with empty liposomes and proteoliposomes.

- Typo: Legend to Figure S4F (line 335, concentration)

We corrected this typo.

Reviewer #2 (Remarks to the Author):

The manuscript by Ballweg et al. has been significantly improved. Considering that the original version was already an great paper, this is quite impressive. Frankly, the authors have gone far beyond what was requested and I recommend acceptance of the manuscript in its present form. The concept that cellular membrane property sensors might be measuring local lipid atom packing densities within the bilayer is novel and could serve as a starting point for a new way of analyzing biological membranes.

We would like to acknowledge the reviewer for the positive assessment. We agree that our new approach to represent MD simulation data might provide new insight into how lipids as a collective affect membrane protein structure and function.

A few minor recommendations:

- Please check the semantics of the sentence about lipid diffusion coefficients in line 268-270. Currently this seems to state that lipid diffusion in model membranes is an order of magnitude lower than in cellular membranes.

Thank you very much for spotting this. We corrected this mistake.

- Remove typos in line 389

We corrected this.

- "points" in line 544 as to be "point"

We corrected this typo.

- line 548: should be "has a significant impact on the structural dynamics..."

We corrected this.

Reviewer #3 (Remarks to the Author):

I think the authors have provided reasonably satisfactory response to the comments. Also, performed additional experiments/simulations to establish their claims. There are some grey areas but given the challenge at hand and the significance of the results I would be happy to accept this manuscript in Nature Communications. I hope this work generates further interest in this important field and leads to deeper understanding of membrane biophysics of sensing of membrane fluidity and lipid saturation leading to various signalling pathways for cellular response to external stress.

Thank you very much for the positive assessment.